# General bounds on the quality of Bayesian coresets

**Trevor Campbell**[*]
Department of Statistics
University of British Columbia
trevor@stat.ubc.ca

## Abstract

Bayesian coresets speed up posterior inference in the large-scale data regime by approximating the full-data log-likelihood function with a surrogate log-likelihood based on a small, weighted subset of the data. But while Bayesian coresets and methods for construction are applicable in a wide range of models, existing theoretical analysis of the posterior inferential error incurred by coreset approximations only apply in restrictive settings—i.e., exponential family models, or models with strong log-concavity and smoothness assumptions. This work presents general upper and lower bounds on the Kullback-Leibler (KL) divergence of coreset approximations that reflect the full range of applicability of Bayesian coresets. The lower bounds require only mild model assumptions typical of Bayesian asymptotic analyses, while the upper bounds require the log-likelihood functions to satisfy a generalized subexponentiality criterion that is weaker than conditions used in earlier work. The lower bounds are applied to obtain fundamental limitations on the quality of coreset approximations, and to provide a theoretical explanation for the previously-observed poor empirical performance of importance sampling-based construction methods. The upper bounds are used to analyze the performance of recent subsample-optimize methods. The flexibility of the theory is demonstrated in validation experiments involving multimodal, unidentifiable, heavy-tailed Bayesian posterior distributions.

## 1 Introduction

Large-scale data is now commonplace in scientific and commerical applications of Bayesian statistics. But despite its prevalence, and the corresponding wealth of research dedicated to scalable Bayesian inference, there are still suprisingly few general methods that provably provide inferential results, within some reasonable tolerated error, at a significant computational cost savings. Exact Markov chain Monte Carlo (MCMC) methods require many full passes over the data [1, Ch. 6–12, 2, Ch. 11–12], limiting the utility of these methods when even a single pass is expensive. A wide range of MCMC methods that access only a subset of data per iteration, e.g., via delayed acceptance [3–6], pseudomarginal or auxiliary variable methods [7–9], and basic subsampling [10–13], provide at most a minor improvement over full-data MCMC [14–16]. On the other hand, methods including carefully constructed log-likelihood function control variates can provide substantial gains [17–19]. However, black-box control variate constructions for large-scale data often rely on assumptions such as posterior density differentiability and unimodality that do not hold in many popular models, e.g., those with discrete variables or multimodality. See [15, 20] for a survey of scalable MCMC methods. Parametric approximations via variational inference [21] or the Laplace approximation [22, 23] can be obtained scalably using stochastic optimization methods, but existing general theoretical guarantees for these methods again typically rely on posterior normality assumptions [24, p. 141–144,25–30] (see [21, 31] for a review).

---

[*] https://trevorcampbell.me

38th Conference on Neural Information Processing Systems (NeurIPS 2024).

Although many existing methods rely on asymptotic normality or unimodality in the large-scale data regime, the problem of handling large-scale data in Bayesian inference does not fundamentally require this structure. Instead, one can more generally exploit *redundancy* in the data (i.e., the existence of good approximate sufficient statistics), which can be used to draw principled conclusions about a large data set based only on a small fraction of examples. Indeed, while approximate posterior normality often does not hold in models with latent discrete or combinatorial objects, weakly identifiable or unidentifiable parameters, persisting heavy tails, multimodality, etc., such models can and regularly do exhibit significant redundancy in the data that can be exploited for faster large-scale inference. *Bayesian coresets* [32]—which involve replacing the full dataset during inference with a sparse weighted subset—are based on this notion of exploiting data redundancy. Empirical studies have shown the existence of high-quality coreset posterior approximations constructed from a small fraction of the data, even in models that violate posterior normality assumptions and for which standard control variate techniques work poorly [33–37]. However, existing theoretical support for Bayesian coresets in the literature is limited. There exist no lower bounds on Bayesian coreset approximation error, and while upper bounds do exist, they currently impose restrictive assumptions. In particular, the best available theoretical upper bounds to date apply to exponential family models [36, 38] and models with strongly log-concave and locally smooth log-densities [37].

This article presents new theoretical techniques and results regarding the quality of Bayesian coreset approximations. The main results are two general large-data asymptotic lower bounds on the KL divergence (Theorems 3.3 and 3.5), as well as a general upper bound on the KL divergence (Theorem 5.3) under the assumption that the log-likelihoods satisfy a multivariate generalization of subexponentiality (Definition 5.2). The main general results in this paper lead to various novel insights about specific Bayesian coreset construction methods. Under mild assumptions,

- common importance-weighted coreset constructions (e.g. [32]) require a coreset size $M$ proportional to the dataset size $N$ (Corollary 4.1), even with post-hoc optimal weight scaling (Corollary 4.2), and thus yield a negligible improvement over full-data inference;

- *any* construction algorithm requires a coreset size $M > d$ when the log-likelihood function is determined by $d$ parameters locally around a point of concentration (Corollary 4.3);

- subsample-optimize coreset construction algorithms (e.g. [36–39]) achieve an asymptotically bounded error with a coreset size $\text{polylog} N$ in a wide variety of models (Corollary 6.1).

The paper includes empirical validation of the main theoretical claims on two models that violate common assumptions made in the literature: a multimodal, unidentifiable Cauchy location model with a heavy-tailed prior, and an unidentifiable logistic regression model with a heavy-tailed prior and persisting posterior heavy tails. Experiments were performed on a computer with an Intel Core i7-8700K and 32GB of RAM. Proofs of all theoretical results may be found in Appendix A.

**Notation.** We use standard asymptotic growth symbols $O, \Omega, \Theta, o, \omega$ (see, e.g., [40, Sec. 3.3]), and their probabilistic variants $O_p, \Omega_p, \Theta_p, o_p, \omega_p$ (see, e.g., [24, Sec. 2.2]). We use the same symbol to denote a measure $\pi$ and its density $\pi(\cdot)$ with respect to a specified dominating measure. We also regularly suppress integration variables and differential symbols in integrals throughout for notational brevity when these are clear from context; for example, $\int \pi \exp(\ell)$ is shorthand for $\int \pi(\mathrm{d}\theta) \exp(\ell(\theta))$. Finally, the pushforward of a measure $\pi$ by a map $\eta$ is denoted simply $\eta\pi$.

## 2   Background

Define a target probability distribution $\pi$ on a space $\Theta$ comprised of a sum of $N$ potentials $\ell_n : \Theta \to \mathbb{R}$, $n = 1, \ldots, N$ and a base distribution $\pi_0(\mathrm{d}\theta)$,

$$\pi(\mathrm{d}\theta) = \frac{1}{Z} \exp\left(\ell(\theta)\right) \pi_0(\mathrm{d}\theta), \qquad \ell(\theta) = \sum_{n=1}^{N} \ell_n(\theta), \qquad \theta \in \Theta,$$

where the normalization constant $Z$ is not known. In the Bayesian context, this distribution corresponds to a Bayesian posterior distribution for a statistical model with prior $\pi_0$ and conditionally i.i.d. data $X_n$, where $\ell_n(\theta) = \log p(X_n|\theta)$. The goal is to compute or approximate expectations under $\pi$; but the likelihood $\ell$ (and its gradient) becomes expensive to evaluate when $N$ is large. To

avoid this cost, *Bayesian coresets* [32–37] involve replacing the target with a surrogate density

$$\pi_w(\mathrm{d}\theta) = \frac{1}{Z(w)} \exp\left(\ell_w(\theta)\right) \pi_0(\mathrm{d}\theta), \qquad \ell_w(\theta) = \sum_{n=1}^{N} w_n \ell_n(\theta), \qquad \theta \in \Theta,$$

where $w \in \mathbb{R}^N$, $w \geq 0$ are a set of weights, and $Z(w)$ is the new normalizing constant. If $w$ has at most $M \ll N$ nonzeros, the $O(M)$ cost of evaluating $\sum_n w_n \ell_n$ (and its gradient) is a significant improvement upon the original $O(N)$ cost. In this work, the problem of coreset construction is formulated in the data-asymptotic limit; a coreset construction method should

- run in $o(N)$ time and memory (or at most $O(N)$ with a small leading constant),
- produce a small coreset of size $M = o(N)$,
- produce a coreset with $O(1)$ posterior forward/reverse KL divergence as $N \to \infty$.

These three desiderata ensure that the effort spent constructing and sampling from the coreset posterior is worthwhile: the coreset provides a meaningful reduction in computational cost compared with standard Markov chain Monte Carlo algorithms, and has a bounded approximation error.

## 3   Lower bounds on approximation error

This section presents lower bounds on the KL divergence of coreset approximations for general models and data generating processes. The first key steps in the analysis are to write all expectations in terms of distributions that do not depend on $w$, and to remove the difficult-to-control influence of the tails of $\pi$ and $\pi_w$ by restricting certain integrals to some small subset $B \subseteq \Theta$ of the parameter space. Lemma 3.1, the key theoretical tool used in this section, achieves both of these two goals; note that the result has no major assumptions and applies generally in any setting that a Bayesian coreset can be used. For convenience, define

$$\underline{\mathrm{KL}}(w) := \min\{\mathrm{KL}(\pi_w||\pi), \mathrm{KL}(\pi||\pi_w)\},$$

and the decreasing, nonnegative function $f : \mathbb{R}_+ \to \mathbb{R}_+$,

$$f(x) = \begin{cases} -\log x + x - 1 & 0 \leq x \leq 1 \\ 0 & x > 1. \end{cases}$$

**Lemma 3.1** (Basic KL Lower Bound). *For all measurable $B \subseteq \Theta$ and coreset weights $w$,*

$$\underline{\mathrm{KL}}(w) \geq f(J_B(w)) \geq 0,$$

*where*

$$J_B(w) = \frac{\int_B \pi_0 \exp \frac{1}{2}(\ell + \ell_w)}{\sqrt{\int \pi_0 \exp(\ell) \int \pi_0 \exp(\ell_w)}} + \sqrt{\pi(B^c)}.$$

Note that while the integrals in the fraction denominator in $J_B(w)$ range over the whole $\Theta$ space, a further lower bound on $\underline{\mathrm{KL}}(w)$ can be obtained by restricting their domains arbitrarily. Also, crucially, the bound in Lemma 3.1 does not depend on $\pi_w(B^c)$, which would be difficult to analyze without detailed knowledge of the tail behaviour of $\pi_w$ as a function of the coreset weights $w$. Although the bound in Lemma 3.1 applies generally, it is most useful when $B$ is small (so that simple local approximations of $\ell$ and $\ell_w$ can be used), $\pi$ concentrates on $B$ (so that $\pi(B^c) \approx 0$), and $\pi$ and $\pi_w$ are very different when restricted to $B$; the behaviour of the bound in this case is roughly (see the proof in Appendix A) $f(J_B(w)) \approx -\log(1 - \mathrm{TV}(\pi, \pi_w))$. Finally, note that Lemma 3.1 remains valid if one replaces $\ell_w$ with $\ell_w - c$ and $\ell$ with $\ell - c'$ for any constants $c, c'$ that do not depend on $\theta$ but may depend on the data and coreset weights $w$.

For the remainder of this section, consider the setting where $\Theta$ is a measurable subset of $\mathbb{R}^d$ for some $d \in \mathbb{N}$, fix some $\theta_0 \in \Theta$, and assume each $\ell_n$ is differentiable in a neighbourhood of $\theta_0$. Let

$$\overline{w} = \sum_n w_n \qquad g = \nabla \ell(\theta_0) \qquad g_w = \nabla \ell_w(\theta_0).$$

Theorems 3.3 and 3.5 characterize KL divergence lower bounds in terms of the sum of the coreset weights $\overline{w}$ and the log-likelihood gradients $g, g_w$. Intuitively for the full data set where all $w_n = 1$ and $\overline{w} = N$, and an i.i.d. data generating process from the likelihood with parameter $\theta_0$, the central limit theorem asserts under mild conditions that $g_w / \overline{w} \overset{p}{\to} 0$ at a rate of $N^{-1/2}$. Theorems 3.3 and 3.5 below provide KL lower bounds when the coreset construction algorithm does not match this behavior. In particular, Theorem 3.3 provides results that are useful when $g_w / \overline{w} \overset{p}{\to} 0$ occurs reasonably quickly but slower than $N^{-1/2}$, while Theorem 3.5 strengthens the conclusion when $g_w / \overline{w} \overset{p}{\to} 0$ very slowly or not at all. The major benefit of Theorems 3.3 and 3.5 for analyzing coreset construction methods is that they reduce the problem of analyzing posterior KL divergence to the much easier problem of analyzing the 2-norm $\| \cdot \|_2$ of a weighted sum of random vectors in $\mathbb{R}^d$.

Consider a sequence $r \to 0$ as $N \to \infty$, and for a fixed matrix $H \succ 0$ let

$$B = \{\theta : (\theta - \theta_0)^T H (\theta - \theta_0) \leq r^2\}$$

be a sequence of neighbourhoods around $\theta_0$; these will appear in Assumptions 3.2 and 3.4 and Theorems 3.3 and 3.5 below. Note that throughout, all asymptotics will be taken as $N \to \infty$, and various sequences (e.g., $r$ and $B$) are implicitly indexed by $N$. To simplify notation, this dependence is suppressed. Assumption 3.2 makes some weak assumptions about the model and data generating process: it intuitively asserts that the potential functions are sufficiently smooth around $\theta_0$, that $r \to 0$ slowly, and that $\pi$ concentrates at $\theta_0$ at a usual rate. Note that Assumption 3.2 does not assume data are generated i.i.d. and places no conditions on the coreset construction algorithm.

**Assumption 3.2.** $\pi_0$ *has a density with respect to the Lebesgue measure,* $\pi_0(\theta_0) > 0$*, each* $\ell_n(\theta)$ *and* $\pi_0(\theta)$ *are twice differentiable in $B$ for sufficiently large $N$, and*

$$\sup_{\theta \in B} \left\| -\frac{1}{N} \nabla^2 \ell(\theta) - H \right\|_2 = o_p(1), \quad \left\| \frac{g}{N} \right\|_2 = O_p\left(N^{-1/2}\right), \quad Nr^2 = \omega(1).$$

Two additional assumptions related to the coreset construction algorithm—namely, that it works well enough that $\frac{1}{\overline{w}} \sum_n w_n \nabla^2 \ell_n(\theta) \overset{p}{\to} H$ and $g_w / \overline{w} \overset{p}{\to} 0$ at a rate faster than $r \to 0$—lead to asymptotic lower bounds on the best possible quality of coresets produced by the algorithm, as well as lower bounds even after optimal post-hoc scaling of the weights.

**Theorem 3.3.** *Suppose Assumption 3.2 holds. If*

$$\sup_{\theta \in B} \left\| -\frac{1}{\overline{w}} \nabla^2 \ell_w(\theta) - H \right\|_2 = o_p(1), \quad \left\| \frac{g_w}{\overline{w}} \right\|_2 = o_p(r),$$

*then*

$$\underline{\mathrm{KL}}(w) \geq O_p(1) + \Omega_p(1) \min\left\{ -\log \pi(B^c), \frac{N\overline{w}}{N + \overline{w}} \left\| \frac{g}{N} - \frac{g_w}{\overline{w}} \right\|_2^2 + d \log \frac{(N + \overline{w})^2}{N \max\{\overline{w}, 1/r^2\}} \right\}$$

$$\min_{\alpha \geq 0} \underline{\mathrm{KL}}(\alpha w) \geq O_p(1) + \Omega_p(1) \min\left\{ -\log \pi(B^c), d \log \left( N \left\| \frac{g}{N} - \frac{g_w}{\overline{w}} \right\|_2^2 \right) \right\}.$$

Theorem 3.3 is restricted to the case where the coreset algorithm is performing reasonably well. Theorem 3.5 extends the bounds to the case where the algorithm is performing poorly, in the sense that it is unable to make $\frac{g_w}{\overline{w}} \overset{p}{\to} 0$ at a rate faster than $r \to 0$ (or perhaps $\frac{g_w}{\overline{w}}$ does not converge to 0 at all). In order to draw conclusions in this setting, we need a weak global assumption on the potential functions. A function $f : \Theta \to \mathbb{R}$ is *L-smooth below* at $\theta_0$ if

$$\forall \theta \in \Theta, \quad f(\theta) \geq f(\theta_0) + \nabla f(\theta_0)^T (\theta - \theta_0) - \frac{L}{2} \|\theta - \theta_0\|_2^2. \tag{1}$$

Note that $L$-smoothness below is weaker than Lipschitz smoothness and does not imply concavity; Eq. (1) restricts the growth of the function only in the negative direction, and only when the expansion is taken at $\theta_0$. Assumption 3.4 asserts that the potential functions are smooth below.

**Assumption 3.4.** *There exist* $L_0, \ldots, L_N, L > 0$ *such that* $\log \pi_0$ *is* $L_0^2$*-smooth below at* $\theta_0$*, for each* $n \in [N]$ $\ell_n$ *is* $L_n^2$*-smooth below at* $\theta_0$*, and* $\frac{1}{N} \sum_{n=1}^{N} L_n^2 \overset{p}{\to} L^2$.

Theorem 3.5 uses Assumptions 3.2 and 3.4 and additional assumptions related to the coreset construction algorithm to obtain lower bounds in a setting that relaxes the "performance" conditions in Theorem 3.3: $-\frac{1}{\overline{w}}\sum_n w_n \nabla^2 \ell_n(\theta)$ no longer needs to converge to $H$ in probability, and $g_w/\overline{w}$ can converge to 0 slowly or not at all.

**Theorem 3.5.** *Suppose Assumptions 3.2 and 3.4 hold. If there exist $\alpha, \beta > 0$ such that*

$$\mathbb{P}\left(\forall \theta \in B, \ -\frac{1}{\overline{w}}\nabla^2 \ell_w(\theta) \succeq \alpha H\right) \to 1, \quad \mathbb{P}\left(\frac{1}{\overline{w}}\sum_n w_n L_n^2 \le \beta L^2\right) \to 1, \quad \left\|\frac{g_w}{\overline{w}}\right\| = \omega_p(r),$$

*then*

$$\underline{\mathrm{KL}}(w) \ge O_p(1) + \Omega_p(1)\min\left\{-\log \pi(B^c), d\log\left(N\min\left\{\left\|\frac{g_w}{\overline{w}}\right\|^2, 1\right\}\right)\right\}.$$

An important final note in this section is that while Theorems 3.3 and 3.5, as stated, require choosing $\Theta$ to be some measurable subset of $\mathbb{R}^d$ and that the posterior $\pi$ concentrates around some point of interest $\theta_0 \in \mathbb{R}^d$, these results can be generalized to a wider class of models and spaces. In particular, Corollary 3.6 demonstrates that if $\Theta$ is arbitrary, but the potential functions $\ell_n$ only depend on $\theta$ through some other function $\eta : \Theta \to \mathbb{R}^d$, that the conclusions of Theorems 3.3 and 3.5 still hold.

**Corollary 3.6.** *Suppose $\Theta$ is an arbitrary measurable space, and the potential functions take the form $\ell_n(\eta(\theta))$ for some measurable function $\eta : \Theta \to \mathbb{R}^d$. Then if the assumptions of Theorems 3.3 and 3.5 hold for potentials $(\ell_n)_{n=1}^N$ as functions on $\mathbb{R}^d$ and pushforward prior $\eta\pi_0$ on $\mathbb{R}^d$, the stated lower bounds also hold for $\min\{\mathrm{KL}(\pi||\pi_w), \mathrm{KL}(\pi_w||\pi)\}$.*

## 4 Lower bound applications

In this section, the general theoretical results from Section 3 are applied to specific algorithms, Bayesian models, and data generating processes to explain previously observed empirical behaviour of coreset construction, as well as to place fundamental limits on the necessary size of coresets. Consider a setting where the data $X_n$ arise as an i.i.d. sequence drawn from some probability distribution $\nu$, $\ell_n(\eta(\theta)) = \log p(X_n|\eta(\theta))$ for $\eta : \Theta \to \mathbb{R}^d$, $\eta_0 = \eta(\theta_0)$, and the following technical criteria hold (where $\mathbb{E}$ denotes expectation under the data generating process):

(A1) $\mathbb{E}\left[\nabla \ell_n(\eta_0)\right] = 0$ and $H = \mathbb{E}\left[-\nabla^2 \ell_n(\eta_0)\right] = \mathbb{E}\left[\nabla \ell_n(\eta_0)\nabla \ell_n(\eta_0)^T\right] \succ 0$.

(A2) $\mathbb{E}\left[\|\nabla \ell_n(\eta_0)\|_2^{2+\delta}\right] < \infty$ for some $\delta > 0$ and $\mathbb{E}\left[\|\nabla^2 \ell_n(\eta_0)\|_F^2\right] < \infty$.

(A3) On a neighbourhood of $\eta_0$, $\|\nabla^2 \ell_n(\eta) - \nabla^2 \ell_n(\eta_0)\|_2 \le R(X_n)\|\eta - \eta_0\|_2$, $\mathbb{E}\left[R(X_n)\right] < \infty$.

(A4) $\eta\pi_0$ is twice differentiable a neighbourhood of $\eta_0$, and $\pi(\eta_0) > 0$.

(A5) For all $r \to 0$ such that $r^2 = \omega(\log N/N)$, $-\log \eta\pi(\|\eta - \eta_0\| > r) = \Omega_p(Nr^2)$.

These conditions apply to a wide range of models, e.g., an unidentifiable, multimodal location model posterior with heavy tails on $\Theta = \mathbb{R}$, where the Bayesian model is specified by

$$\theta \sim \mathrm{Cauchy}(0,1) \qquad\qquad (X_n)_{n=1}^N \overset{\text{iid}}{\sim} \mathrm{Cauchy}(\theta^2, 1), \qquad\qquad (2)$$

and the data are generated from the likelihood with parameter $\theta_0 = 5$, and an unidentifiable logistic regression posterior with heavy tails on $\mathbb{R}^2$, where the Bayesian model is specified by

$$\theta \sim \mathrm{Cauchy}(0, I) \qquad Y_n \overset{\text{ind}}{\sim} \mathrm{Bern}\left(\frac{1}{1 + e^{-X_n^T A\theta}}\right) \qquad A = \begin{bmatrix} 1 & 1 \\ 1 & 1 \end{bmatrix}, \qquad (3)$$

the covariates are generated via $X_n \overset{\text{iid}}{\sim} \mathrm{Unif}(\{x \in \mathbb{R}^2 : \|x\|_2 \le 1\})$, and the observations $Y_n$ are generated from the likelihood with parameter $\theta_0 = [1 \quad 6]^T$. See Proposition A.6 in Appendix A for the verification of (A1-5) for these two models. Example posterior log-densities for these models are displayed in Fig. 1.

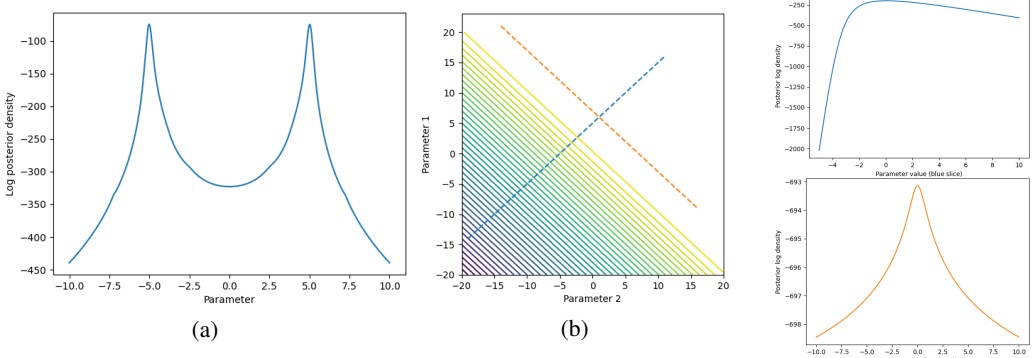

Figure 1: Example unnormalized posterior densities given 50 data points for (1a) the Cauchy location model and (1b) the logistic regression model. The orange and blue dashed lines in (1b) indicate one-dimensional slices that are shown in the rightmost panels.

---

**Algorithm 1** Importance-weighted coreset construction

---

Compute probabilities $(p_n)_{n=1}^N$ (may depend on the data and model)
Draw $I_1, \ldots, I_M \overset{\text{iid}}{\sim} \text{Categorical}(p_1, \ldots, p_N)$
For each $n$, set $w_n = \frac{1}{Mp_n} \sum_{m=1}^M \mathbb{1}[I_m = n]$.
**return** $(w_n)_{n=1}^N$

---

**Algorithm 2** Scaled importance-weighted coreset construction

---

Obtain coreset weights $(w_n)_{n=1}^N$ via Algorithm 1
Compute $\alpha^\star = \arg\min_{\alpha \geq 0} \text{KL}(\pi_{\alpha w} || \pi)$
**return** $(\alpha^\star w_n)_{n=1}^N$

---

### 4.1 Minimum coreset size for importance-weighted coresets

A popular algorithm for coreset construction that has appeared in a wide variety of domains—e.g., Bayesian inference [32, 33, Section 4.1], frequentist inference (e.g., [41–45]), and optimization (see [46] for a recent survey)—involves subsampling of the data followed by an importance-weighting correction. The pseudocode is given in Algorithm 1. Note that $\mathbb{E}[w_n] = 1$, and so $\mathbb{E}[\ell_w] = \ell$; the coreset potential is an unbiased estimate of the exact potential. The advantage of this method is that it is straightforward and computationally efficient. If the sampling probabilities are uniform $p_n = 1/N$, then Algorithm 1 constructs a coreset in $O(M)$ time and $O(M)$ memory. Nonuniform probabilities $p_n$ require $\Omega(N)$ time, as they require a pass over all $N$ data points to compute each $p_n$ [32, 42] followed by sampling the coreset, e.g., via an alias table [47, 48]. However, empirical results produced by this methodology have generally been underwhelming, even with carefully chosen sampling probabilities; see, e.g., Figure 2 of [32].

Corollary 4.1 explains these poor results: Bayesian coresets constructed via Algorithm 1 must satisfy $M \propto N$ in order to maintain a bounded $\underline{\text{KL}}(w)$ in the data-asymptotic limit. In other words, such coresets do not satisfy the desiderata in Section 2. The only restriction is that there exist constants $c, C > 0$ such that for all $N \in \mathbb{N}$, the sampling probabilities $(p_n)_{n=1}^N$ satisfy

$$(\text{A6}) \qquad 0 < c \leq \min_n Np_n \leq \max_n Np_n \leq C < \infty \quad a.s. \tag{4}$$

The lower threshold ensures that the variance of the importance-weighted log-likelihood is not too large, while the upper threshold ensures sufficient diversity in the draws from subsampling. The condition in Eq. (4) is not a major restriction, in the sense that performance should deteriorate even further when it does not hold. The $(p_n)_{n=1}^N$ may otherwise depend arbitrarily on the data and model.

**Corollary 4.1.** *Given (A1-6), $M \to \infty$, and $M = o(N)$, coresets produced by Algorithm 1 satisfy*

$$\underline{\text{KL}}(w) = \Omega_p\left(\frac{N}{M}\right). \tag{5}$$

The intuition behind Corollary 4.1 is that both the true posterior and the importance-weighted coreset posterior are asymptotically approximately normal with variance $\propto 1/N$ as $N \to \infty$; however, the coreset posterior mean is roughly $\propto M^{-1/2}$ away from the posterior mean, because the subsample is of size $M$. The KL divergence between two Gaussians is lower-bounded by the inverse variance times the mean difference squared, yielding $\approx N/M$ as in Eq. (5).

Given the intuition that the coreset posterior mean is far from the posterior mean relative to their variances, it is worth asking whether one can apply a small amount of effort to "correct" the importance-weighted coreset by scaling the weights (and hence the variance) down, as shown in Algorithm 2. Unfortunately, Corollary 4.2 demonstrates that even with optimal scaling, $M \propto N$ is still required in order to maintain a bounded KL divergence as $N \to \infty$.

**Corollary 4.2.** *Given (A1-6), $M \to \infty$, and $M = o(N)$, coresets produced by Algorithm 1 satisfy*

$$\min_{\alpha > 0} \underline{\mathrm{KL}}(\alpha w) = \Omega_p \left( \log \frac{N}{M} \right).$$

Fig. 2 provides empirical confirmation of Corollaries 4.1 and 4.2 on the Cauchy location and logistic regression models in Eqs. (2) and (3). In particular, these figures show that the empirical rates of growth of KL as a function of $N$ closely matches $\Omega_p(\frac{N}{M})$ for importance-weighted coresets, and $\Omega_p(\log \frac{N}{M})$ for the same with post-hoc scaling, for a wide range of coreset sizes $M \in \{\log N, \sqrt{N}, 1/2N\}$. Thus, importance weighted coreset construction methods do not satisfy the desiderata in Section 2 for a wide range of models, and alternate methods should be considered.

## 4.2 Minimum coreset size for any coreset construction

This section extends the minimum coreset size results from importance-weighted schemes to *any* coreset construction algorithm. In particular, Corollary 4.3 shows that under (A7)—a strengthening of (A3) and Assumption 3.4—and (A8)—which asserts that $\nabla \ell_1(\eta_0), \ldots, \nabla \ell_M(\eta_0)$ are linearly independent a.s. and satisfy a technical moment condition—at least $d$ coreset points are required to keep the KL divergence bounded as $N \to \infty$.

(A7) Assumption 3.4 holds and there exists $\gamma > 0$ such that for all sufficiently large $N \in \mathbb{N}$,

$$\forall \eta \in B, n \in [N], \quad -\nabla^2 \ell_n(\eta) \succeq \gamma H \quad \text{and} \quad L_n^2 < \gamma^{-1} L^2.$$

(A8) For all coreset sizes $M < d$, there exists a $\delta > 0$ such that

$$\mathbb{E}\left[\left(1^T (G^T G)^{-1} 1\right)^{M+\delta}\right] < \infty \qquad G = [\nabla \ell_1(\eta_0) \quad \ldots \quad \nabla \ell_M(\eta_0)] \in \mathbb{R}^{d \times M}.$$

**Corollary 4.3.** *For a fixed coreset size $M < d$, given (A1-5,7,8),*

$$\min_{w \in \mathbb{R}_+^N : \|w\|_0 \leq M} \underline{\mathrm{KL}}(w) = \Omega_p(\log N).$$

# 5 Upper bounds on approximation error

This section presents upper bounds on the KL divergence of coreset approximations. As in Section 3, the first step is to write all expectations in terms of distributions that do not depend on $w$. Lemma 5.1 does so without imposing any major assumptions; the result again applies generally in any setting that a Bayesian coreset can be used. For convenience, define

$$\overline{\mathrm{KL}}(w) := \max\{\mathrm{KL}(\pi_w \| \pi), \mathrm{KL}(\pi \| \pi_w)\}.$$

**Lemma 5.1** (Basic KL Upper Bound). *For all coreset weights $w$,*

$$\overline{\mathrm{KL}}(w) \leq \inf_{\lambda > 0} \frac{1}{\lambda} \log \int \pi \exp\left((1+\lambda)(\bar{\ell}_w - \bar{\ell})\right),$$

*where for all $n \in [N]$, $\bar{\ell}_n = \ell_n - \int \pi \ell_n$, $\bar{\ell} = \sum_n \bar{\ell}_n$, and $\bar{\ell}_w = \sum_n w_n \bar{\ell}_n$.*

The upper bound in Lemma 5.1 is nonvacuous (i.e., finite) as long as there exists a $\alpha > 1$ such that the $\alpha$ Rényi divergence $D_\alpha(\pi_w \| \pi)$ [49, p. 3799] is finite. Note that as in Lemma 3.1, the bound in

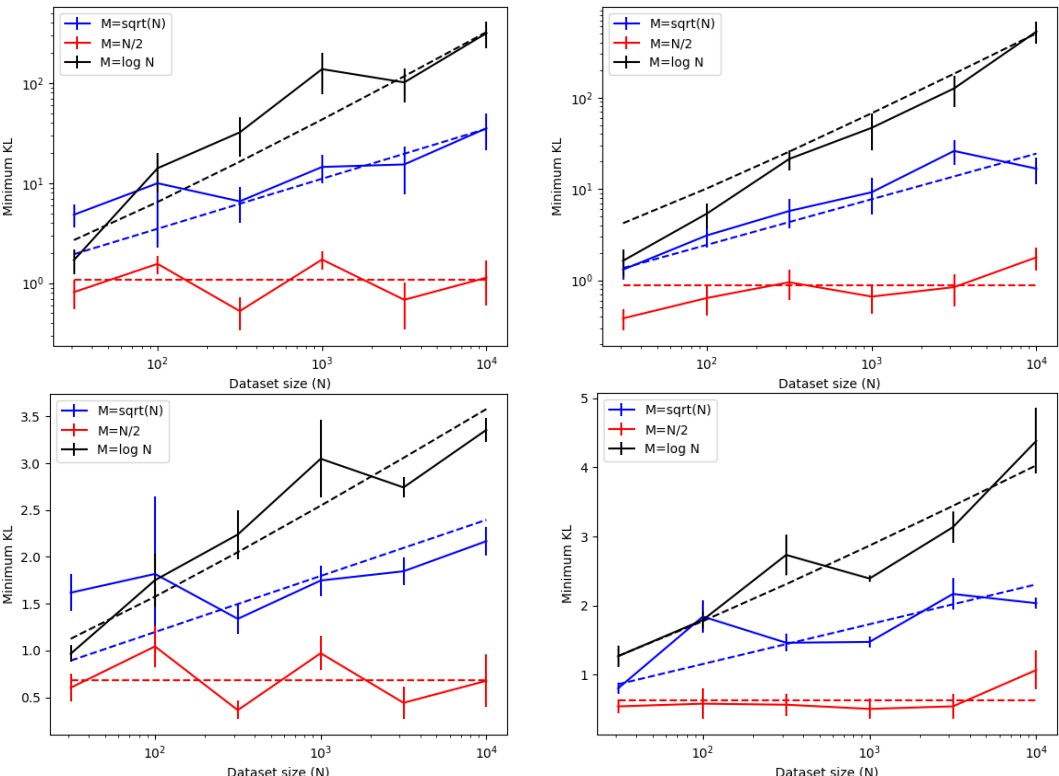

Figure 2: Importance-weighted coreset quality, showing the minimum of the forward and reverse KL divergences on the vertical axis as a function of dataset size $N$ for 3 coreset sizes: $\log N$ (black), $\sqrt{N}$ (blue), and $1/2N$ (red). Dashed lines indicate predictions from the theory in Corollaries 4.1 and 4.2, solid lines indicate the mean over 10 trials, and error bars indicate standard error. The top row shows the quality of basic importance-weighted coresets (note that both horizontal and vertical axes are in log scale), while the bottom row shows the quality with optimal post-hoc scaling (note that only the horizontal axis is in log scale). The left column corresponds to the Cauchy location model, while the right column corresponds to the logistic regression model. Sampling probabilities $p_n$ for both models are set proportional to $X_n^2$, thresholded to lie between $0.1/N$ and $10/N$.

Lemma 5.1 remains valid if one replaces $\ell_w$ with $\ell_w - c$ and $\ell$ with $\ell - c'$ for any constants $c, c'$ that do not depend on $\theta$ but may depend on the coreset weights $w$ and data.

More practical bounds necessitate an assumption about the behaviour of the potentials $(\ell_n)_{n=1}^N$. Definition 5.2 below asserts that the multivariate moment generating function of $(\ell_n)_{n=1}^N$ is bounded when the vector is close to 0. This definition is a generalization of the usual definition of subexponentiality for the univariate setting (e.g., [50, Sec. 2.7]). Theorem 5.3 subsequently shows that Definition 5.2 is sufficient to obtain simple bounds on $\overline{\mathrm{KL}}$.

**Definition 5.2.** *For $A \in \mathbb{R}^{N \times N}$, $A \succeq 0$, and monotone function $h : \mathbb{R}_+ \to \mathbb{R}_+$ such that $\lim_{x \to 0} h(x) = h(0) = 0$, the potentials $(\ell_n)_{n=1}^N$ are $(h, A)$-subexponential if*

$$\forall w \in \mathbb{R}^N : w^T A w \leq 1, \qquad \int \pi \exp(\bar{\ell}_w) \leq \exp(h(w^T A w)).$$

**Theorem 5.3.** *If the potentials $(\ell_n)_{n=1}^N$ are $(h, A)$-subexponential, then*

$$\forall w \in \mathbb{R}_+^N : 4(w-1)^T A (w-1) \leq 1, \qquad \overline{\mathrm{KL}}(w) \leq h(4(w-1)^T A (w-1)).$$

Definition 5.2, the key assumption in Theorem 5.3, is satisfied by a wide range of models when choosing $h(x) = x$ and $A \propto \mathrm{Cov}_\pi((\ell_n)_{n=1}^N)$, as demonstrated by Proposition 5.4. Because this case applies widely, let $A$-*subexponential* be shorthand for $(h, A)$-subexponentiality with $h(x) = x$.

**Proposition 5.4.** *If for all $w$ in a ball centered at the origin, $\int \pi \exp(\bar{\ell}_w) < \infty$, then there exists $\beta > 0$ such that the potentials $(\ell_n)_{n=1}^N$ are $\beta \mathrm{Cov}_\pi((\ell_n)_{n=1}^N)$-subexponential.*

---

**Algorithm 3** Subsample-optimize coreset construction

---

Compute probabilities $(p_n)_{n=1}^N$ (may depend on the data and model)

Draw $I_1, \ldots, I_M \overset{\text{iid}}{\sim} \text{Categorical}(p_1, \ldots, p_N)$, and set $\mathcal{I} = \{I_1, \ldots, I_M\}$

Compute $w^\star = \arg\min_{w \in \mathbb{R}_+^N} \text{KL}(\pi_w \| \pi)$   s.t.   $w_n \neq 0$ only if $n \in \mathcal{I}$.

**return** $(w_n^\star)_{n=1}^N$

---

In other words, intuitively, if a coreset construction algorithm produces weights such that $\text{Var}_\pi(\bar{\ell}_w - \bar{\ell})$ is small, then $\overline{\text{KL}}(w)$ is also small. That being said, the generality of Definition 5.2 to allow arbitrary $h, A$ is still helpful in obtaining upper bounds in specific cases; see, e.g., Propositions A.1 and A.2.

## 6  Upper bound application: subsample-optimize coresets

A strategy to construct Bayesian coresets that has recently emerged in the literature, shown in Algorithm 3, is to first subsample the data to select $M$ data points, and subsequently optimize the weights for those selected data points [36–38]. The subsampling step serves to pick a reasonably flexible basis of log-likelihood functions for coreset approximation, and avoids the slow greedy selection routines from earlier work [33–35]. The optimization step tunes the weights for the selected basis, avoiding the poor approximations of importance-weighting methods. Indeed, Algorithm 3 creates exact coresets $\pi_{w^\star} = \pi$ with high probability in Gaussian location models [36, Prop. 3.1] and finite-dimensional exponential family models [37, Thm. 4.1], and near-exact coresets with high probability in strongly log-concave models [37, Thm. 4.2] and Bayesian linear regression [38, Prop. 3].

Corollary 6.1 generalizes these results substantially, and demonstrates that coresets of size $M = O(\text{polylog}(N))$ produced by the subsample-optimize method in Algorithm 3 maintain a bounded KL divergence as $N \to \infty$. Two key assumptions are subexponentiality of the potentials and a polynomial (in $N$) growth of $\text{Var}_\pi(\ell(\theta))$; these conditions are not stringent and should hold for a wide range of Bayesian models and i.i.d. data generating processes. The last key assumption in Eq. (6) is that a randomly-chosen potential function $\ell_I$, $I \sim \text{Categorical}(p_1, \ldots, p_N)$ (with probabilities as in Algorithm 3) is well-aligned with the residual coreset error function. Similar alignment conditions have appeared in past results for more restrictive settings (see, e.g., $J(\delta)$ in [37, Thm. 4.1]).

**Corollary 6.1.** *Suppose there exist $\beta, \alpha > 0$ and $0 \leq \rho, \epsilon < 1$ such that the potential functions $(\ell_n)_{n=1}^N$ are $\beta \, \text{Cov}_\pi((\ell_n)_{n=1}^N)$-subexponential with probability increasing to 1 as $N \to \infty$, $\text{Var}_\pi(\ell(\theta)) = O_p(N^\alpha)$, $M = (\log N)^{\frac{1}{1-\rho}}$, and*

$$\mathbb{P}\Big(\max\{0, \text{Corr}_\pi\big(\ell_{I_M}(\theta), \ell(\theta) - \ell_{M-1}^\star(\theta)\big)\}^2 \geq 1 - \epsilon \Big| (\ell_n)_{n=1}^N\Big) = \omega_p(M^{-\rho}) \quad (6)$$

$$\ell_{M-1}^\star(\theta) = \underset{g \in \text{cone}\{\ell_{I_1}, \ldots, \ell_{I_{M-1}}\}}{\arg\min} \text{Var}_\pi(\ell(\theta) - g(\theta)) \qquad I_1, \ldots, I_M \overset{\text{iid}}{\sim} \text{Categorical}(p_1, \ldots, p_N).$$

*Then Algorithm 3 produces a coreset with $\overline{\text{KL}}(w) = O_p(1)$ as $N \to \infty$.*

Fig. 3 confirms that subsample-optimize coreset construction methods applied to the logistic regression and Cauchy location models in Eqs. (2) and (3) (which both violate the conditions of past upper bounds in the literature) are able to provide high-quality posterior approximations for very small coresets—in this case, $M \propto \log N$.

## 7  Conclusions

This article presented new general lower and upper bounds on the quality of Bayesian coreset approximations, as measured by the KL divergence. These results were used to draw novel conclusions regarding importance-weighted and subsample-optimize coreset methods, which align with simulation experiments on two synthetic models that violate the assumptions of past theoretical results. Avenues for future work include general bounds on the subexponentiality constant $\beta$ in Proposition 5.4, as well as the alignment probability in Eq. (6), in the setting of Bayesian models with i.i.d. data generating processes. A limitation of this work is that both quantities currently require case-by-case analysis.

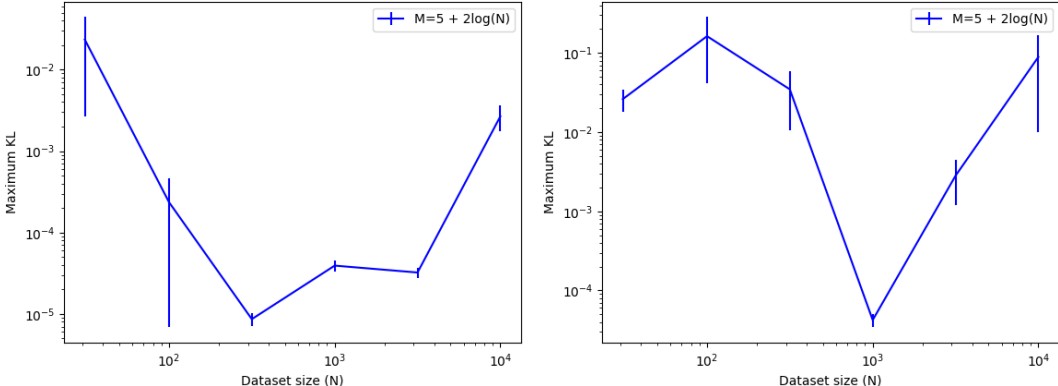

Figure 3: Subsample-optimize coreset quality, showing the maximum of the forward and reverse KL divergences on the vertical axis as a function of dataset size $N$ for coresets of size $5 + 2 \log N$. Solid lines indicate the mean over 70 trials, and error bars indicate standard error. The left panel is for the Cauchy location model, while the right panel is for the logistic regression model. Sampling probabilities are uniform $p_n = 1/N$, and coreset weights were optimized by nonnegative least squares for log-likelihoods discretized via samples from $\pi$ [34, Eq. 4].

## Acknowledgments and Disclosure of Funding

The author gratefully acknowledges the support of an NSERC Discovery Grant (RGPIN-2019-03962).

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

# A  Proofs

*Proof of Lemma 3.1.* By Vajda's inequality [51],

$$\underline{\text{KL}}(w) \geq \log \frac{1 + \text{TV}(\pi, \pi_w)}{1 - \text{TV}(\pi, \pi_w)} - \frac{2\,\text{TV}(\pi, \pi_w)}{1 + \text{TV}(\pi, \pi_w)}$$
$$\geq -\log\left(1 - \text{TV}(\pi, \pi_w)\right) - \text{TV}(\pi, \pi_w)$$
$$\geq 0.$$

The bound is monotone increasing in $\text{TV}(\pi, \pi_w)$; therefore because the squared Hellinger distance satisfies the inequality [52, p. 61],

$$H^2(\pi, \pi_w) = \frac{1}{2} \int \left(\sqrt{\pi} - \sqrt{\pi_w}\right)^2 \leq \frac{1}{2} \int |\pi - \pi_w| = \text{TV}(\pi, \pi_w),$$

we have that

$$\underline{\text{KL}}(w) \geq -\log\left(1 - H^2(\pi, \pi_w)\right) - H^2(\pi, \pi_w).$$

We substitute the value of the squared Hellinger distance to find that

$$\underline{\text{KL}}(w) \geq -\log\left(\int \sqrt{\pi \pi_w}\right) + \int \sqrt{\pi \pi_w} - 1 \geq 0.$$

Note that $\int \sqrt{\pi \pi_w} \leq 1$, so

$$\underline{\text{KL}}(w) \geq -\log\left(\min\{1, \int \sqrt{\pi \pi_w}\}\right) + \min\{1, \int \sqrt{\pi \pi_w}\} - 1 \geq 0.$$

The bound is monotone decreasing in $\int \sqrt{\pi \pi_w}$, so we require an upper bound on $\int \sqrt{\pi \pi_w}$. To obtain the required bound, we split the integral into two parts—one on the set $B$, and the other on $B^c$—and then use the Cauchy-Schwarz inequality to bound the part on $B^c$. Note that by definition $\pi$ and $\pi_w$ are mutually dominating, so the density ratio $\pi_w/\pi$ is well-defined and measurable.

$$\int \sqrt{\pi \pi_w} = \int_B \sqrt{\pi \pi_w} + \int_{B^c} \sqrt{\pi \pi_w}$$
$$= \int_B \sqrt{\pi \pi_w} + \int \pi \sqrt{\frac{\pi_w}{\pi}} \mathbb{1}_{B^c}$$
$$\leq \int_B \sqrt{\pi \pi_w} + \sqrt{\pi(B^c)}$$
$$= \frac{\int_B \pi_0 \exp \frac{1}{2}(\ell + \ell_w)}{\sqrt{\int \pi_0 \exp(\ell) \int \pi_0 \exp(\ell_w)}} + \sqrt{\pi(B^c)}.$$

The result follows. $\qquad\square$

*Proof of Lemma 5.1.* We first consider the forward KL divergence. By definition,

$$\text{KL}(\pi || \pi_w) = \int \pi(\ell - \ell_w) + \log \frac{\int \pi_0 \exp(\ell_w)}{\int \pi_0 \exp(\ell)}$$
$$= \int \pi(\ell - \ell_w) + \log \int \pi \exp(\ell_w - \ell).$$

Since the KL is positive, for $\lambda > 0$,

$$\text{KL}(\pi || \pi_w) \leq \frac{1 + \lambda}{\lambda} \int \pi(\ell - \ell_w) + \frac{1 + \lambda}{\lambda} \log \int \pi \exp(\ell_w - \ell)$$
$$\leq \frac{1 + \lambda}{\lambda} \int \pi(\ell - \ell_w) + \frac{1}{\lambda} \log \int \pi \exp((1 + \lambda)(\ell_w - \ell))$$
$$= \frac{1}{\lambda} \log \int \pi \exp((1 + \lambda)(\bar{\ell}_w - \bar{\ell})),$$

by Jensen's inequality. Next we consider the reverse KL divergence. For any $\lambda \neq 0$,

$$\text{KL}(\pi_w || \pi) = \int \pi_w(\ell_w - \ell) + \log \frac{\int \pi_0 \exp(\ell)}{\int \pi_0 \exp(\ell_w)}$$
$$= \frac{1}{\lambda} \int \pi_w \lambda(\ell_w - \ell) - \log \int \pi \exp(\ell_w - \ell).$$

By Jensen's inequality, for $\lambda > 0$,

$$\mathrm{KL}(\pi_w\|\pi) \le \frac{1}{\lambda}\log\int \pi_w \exp(\lambda(\ell_w - \ell)) - \log\int \pi \exp(\ell_w - \ell)$$

$$= \frac{1}{\lambda}\log\frac{\int \pi \exp((1+\lambda)(\ell_w - \ell))}{\int \pi \exp(\ell_w - \ell)} - \log\int \pi\exp(\ell_w - \ell)$$

$$= \frac{1}{\lambda}\log\int \pi \exp((1+\lambda)(\ell_w - \ell)) - \frac{1+\lambda}{\lambda}\log\int \pi\exp(\ell_w - \ell)$$

$$\le \frac{1+\lambda}{\lambda}\int \pi(\ell - \ell_w) + \frac{1}{\lambda}\log\int \pi\exp((1+\lambda)(\ell_w - \ell))$$

$$= \frac{1}{\lambda}\log\int \pi\exp((1+\lambda)(\bar\ell_w - \bar\ell)).$$

This is the same bound as in the forward KL divergence case. Since the bound applies for all $\lambda > 0$, we can take the infimum. □

*Proof of Theorem 3.3.* By replacing the integrals over the whole space $\Theta$ in the denominator of $J_B(w)$ in Lemma 3.1 with integrals over the subset $B$,

$$\underline{\mathrm{KL}}(w) \ge -\log\min(1, J_B(w)) + \min(1, J_B(w)) - 1$$
$$\ge O_p(1) - \log J_B(w)$$
$$\ge O_p(1) + \min\left\{G_B(w), -\log\sqrt{\pi(B^c)}\right\}$$

$$G_B(w) = -\log\int_B \pi_0\exp((1/2)(\ell + \ell_w)) + \frac{1}{2}\log\int_B \pi_0\exp(\ell) + \frac{1}{2}\log\int_B \pi_0\exp(\ell_w).$$

So to obtain the stated lower bound on the KL divergence, we require an upper bound on $\log\int_B \pi_0\exp((1/2)(\ell + \ell_w))$, and lower bounds on $\log\int_B \pi_0\exp(\ell)$ and $\log\int_B \pi_0\exp(\ell_w)$. By Taylor's theorem, Assumption 3.2, and the assumption on $\nabla^2\ell_w(\theta)$, for all $\theta \in B$,

$$\left|\ell(\theta) - \ell(\theta_0) - g^T(\theta - \theta_0) + \frac{N}{2}(\theta - \theta_0)^T H(\theta - \theta_0)\right| \le \frac{No_p(1)}{2}(\theta - \theta_0)^T H(\theta - \theta_0)$$

$$\left|\ell_w(\theta) - \ell_w(\theta_0) - g_w^T(\theta - \theta_0) + \frac{\bar w}{2}(\theta - \theta_0)^T H(\theta - \theta_0)\right| \le \frac{\bar w o_p(1)}{2}(\theta - \theta_0)^T H(\theta - \theta_0). \tag{7}$$

We shift the exponential arguments in $G_B(w)$ by $(1/2)(\ell(\theta_0) + \ell_w(\theta_0))$, note that $\pi_0$ is continuous and positive around $\theta_0$, and and apply the Taylor expansions in Eq. (7) to obtain an upper bound on the first term:

$$\log\int_B \pi_0 e^{\frac{1}{2}(\ell - \ell(\theta_0) + \ell_w - \ell_w(\theta_0))} \le O_p(1) + \log\int_B e^{\frac{1}{2}((g + g_w)^T(\theta - \theta_0) - \frac{(\sim 1)(N + \bar w)}{4}(\theta - \theta_0)^T H(\theta - \theta_0)}},$$

where $(\sim 1)$ denotes a quantity that converges in probability to 1 as $N \to \infty$. We can transform variables to $x = C^T(\theta - \theta_0)$, where $H = CC^T$ is the Cholesky factorization of $H$, and subsequently complete the square:

$$\log\int_B \pi_0 e^{\frac{1}{2}(\cdots)} \le O_p(1) + \frac{(\sim 1)\|C^{-1}(g + g_w)\|^2}{4(N + \bar w)} + \log\int_{\|x\|^2 \le r^2} e^{-\frac{(\sim 1)(N + \bar w)}{4}\left\|x - \frac{(\sim 1)C^{-1}(g + g_w)}{(N + \bar w)}\right\|^2}. \tag{8}$$

We can obtain lower bounds on the other two terms using a similar technique:

$$\log\int_B \pi_0 e^{\ell - \ell(\theta_0)} \ge O_p(1) + \frac{(\sim 1)\|C^{-1}g\|^2}{2N} + \log\int_{\|x\|^2 \le r^2} e^{-\frac{(\sim 1)N}{2}\left\|x - \frac{(\sim 1)C^{-1}g}{N}\right\|^2} \tag{9}$$

$$\log\int_B \pi_0 e^{\ell_w - \ell_w(\theta_0)} \ge O_p(1) + \frac{(\sim 1)\|C^{-1}g_w\|^2}{2\bar w} + \log\int_{\|x\|^2 \le r^2} e^{-\frac{(\sim 1)\bar w}{2}\left\|x - \frac{(\sim 1)C^{-1}g_w}{\bar w}\right\|^2}. \tag{10}$$

It remains to analyze the three $\log\int\ldots$ terms. We bound the integral term in Eq. (8) with the integral over the whole space:

$$\log\int_{\|x\|^2 \le r^2} e^{-\frac{(\sim 1)(N + \bar w)}{4}\|\cdots\|^2} \le O_p(1) - \frac{d}{2}\log(N + \bar w).$$

For the integral term in Eq. (9), note that since $Nr^2 = \omega(1)$ and $\|C^{-1}g/N\| = O_p(N^{-1/2})$, we have

$$\log\int_{\|x\|^2 \le r^2} e^{-\frac{(\sim 1)N}{2}\|\cdots\|^2}$$

$$= \log\left(\int e^{-\frac{(\sim 1)N}{2}}(\cdots) - \int_{\|x\|^2 > r^2} e^{-\frac{(\sim 1)N}{2}\left\|x - \frac{(\sim 1)C^{-1}g}{N}\right\|^2}\right)$$

$$\geq \log\left(\left(\frac{(\sim 1)2\pi}{N}\right)^{d/2} - e^{-\frac{(\sim 1)N}{4}\min_{\|x\|\geq r}\left\|x - \frac{(\sim 1)C^{-1}g}{N}\right\|^2}\int e^{-\frac{(\sim 1)N}{4}\left\|x - \frac{(\sim 1)C^{-1}g}{N}\right\|^2}\right)$$

$$= \log\left(\left(\frac{(\sim 1)2\pi}{N}\right)^{d/2} - e^{-\frac{\Omega_p(Nr^2)}{4}}\left(\frac{(\sim 1)4\pi}{N}\right)^{d/2}\right)$$

$$= -\frac{d}{2}\log(N) + O_p(1).$$

For the integral term in Eq. (10), we consider two cases: one where $\overline{w}$ is large, and one where it is small. First assume $\overline{w}r^2 > 8d\log 2$; then by a similar technique as used in the first lower bound, since $\|C^{-1}g_w/\overline{w}\| = o_p(r)$,

$$\log\int_{\|x\|^2 \leq r^2} e^{-\frac{(\sim 1)\overline{w}}{2}\|\cdots\|^2}$$

$$\geq \log\left(\left(\frac{(\sim 1)2\pi}{\overline{w}}\right)^{d/2} - e^{-\frac{(\sim 1)\overline{w}}{4}\min_{\|x\|\geq r}\left\|x - \frac{(\sim 1)C^{-1}g_w}{\overline{w}}\right\|^2}\int e^{-\frac{(\sim 1)\overline{w}}{4}\|x - \frac{(\sim 1)C^{-1}g_w}{\overline{w}}\|^2}\right)$$

$$\geq \log\left(\left(\frac{(\sim 1)2\pi}{\overline{w}}\right)^{d/2} - e^{-2d\log 2(\sim 1)}\left(\frac{(\sim 1)4\pi}{\overline{w}}\right)^{d/2}\right)$$

$$\geq -\frac{d}{2}\log\overline{w} + O_p(1).$$

When $\overline{w}r^2 \leq 8d\log 2$, we transform variables $y = x/r$ to find that since $\|C^{-1}g_w/\overline{w}\| = o_p(r)$,

$$\log\int_{\|x\|^2 \leq r^2} e^{-\frac{(\sim 1)\overline{w}}{2}\|\cdots\|^2} = \frac{d}{2}\log r^2 + \log\int_{\|y\|^2 \leq 1} e^{-\frac{(\sim 1)\overline{w}r^2}{2}\left\|y - \frac{(\sim 1)C^{-1}g_w}{r\overline{w}}\right\|^2}$$

$$\geq \frac{d}{2}\log r^2 + \log e^{-\frac{8d\log 2(\sim 1)}{2}\left(2 + 2\left\|\frac{(\sim 1)C^{-1}g_w}{r\overline{w}}\right\|^2\right)}\left(\int_{\|y\|^2 \leq 1} 1\right)$$

$$= \frac{d}{2}\log r^2 + O_p(1).$$

Therefore regardless of the value of $\overline{w}$,

$$\log\int_{\|x\|^2 \leq r^2} e^{-\frac{(\sim 1)\overline{w}}{2}\|\cdots\|^2} \geq -\frac{d}{2}\log\left(\max\{\overline{w}, 1/r^2\}\right) + O_p(1).$$

So therefore combining all previous results,

$$G_B(w) \geq O_p(1) + \frac{(\sim 1)}{4}\left(\frac{\|C^{-1}g\|^2}{N} + \frac{\|C^{-1}g_w\|^2}{\overline{w}} - \frac{\|C^{-1}(g + g_w)\|^2}{N + \overline{w}}\right) + \frac{d}{4}\log\frac{(N + \overline{w})^2}{N\max\{\overline{w}, 1/r^2\}}$$

$$= O_p(1) + \frac{(\sim 1)}{4}\left(\frac{\overline{w}\|C^{-1}g\|^2}{N(N + \overline{w})} + \frac{N\|C^{-1}g_w\|^2}{\overline{w}(N + \overline{w})} - \frac{2g^T H^{-1}g_w}{N + \overline{w}}\right) + \frac{d}{4}\log\frac{(N + \overline{w})^2}{N\max\{\overline{w}, 1/r^2\}}$$

$$= O_p(1) + \frac{(\sim 1)}{4}\left(\frac{N\overline{w}}{N + \overline{w}}\left\|\frac{C^{-1}g}{N} - \frac{C^{-1}g_w}{\overline{w}}\right\|^2\right) + \frac{d}{4}\log\frac{(N + \overline{w})^2}{N\max\{\overline{w}, 1/r^2\}}$$

$$= O_p(1) + \Omega_p(1)\left(\frac{N\overline{w}}{N + \overline{w}}\left\|\frac{g}{N} - \frac{g_w}{\overline{w}}\right\|^2 + d\log\frac{(N + \overline{w})^2}{N\max\{\overline{w}, 1/r^2\}}\right).$$

We now consider the minimum over $\alpha \geq 0$. Since neither $O_p(1)$ or $\Omega_p(1)$ above depends on $\overline{w}$, we have that

$$\min_{\alpha \geq 0}\underline{KL}(\alpha w) \geq O_p(1) + \Omega_p(1)\min\left\{-\log\pi(B^c), \left(\min_{\alpha \geq 0}\frac{N\alpha\overline{w}}{N + \alpha\overline{w}}\left\|\frac{g}{N} - \frac{g_w}{\overline{w}}\right\|^2 + d\log\frac{(N + \alpha\overline{w})^2}{N\max\{\alpha\overline{w}, 1/r^2\}}\right)\right\}.$$

On the $1/r^2$ branch of the objective function, the derivative in $\alpha$ is always positive, and hence the minimum occurs at $\alpha = 0$, and so

$$\min_{\alpha \geq 0}(\dots) \geq d\log(Nr^2).$$

On the $\alpha w$ branch of the objective function,

$$\min_{\alpha \geq 0}\frac{N\alpha\overline{w}}{N + \alpha\overline{w}}\left\|\frac{g}{N} - \frac{g_w}{\overline{w}}\right\|^2 + d\log\frac{(N + \alpha\overline{w})^2}{N\alpha\overline{w}} \geq \min_{\alpha \geq 0}\frac{N\alpha\overline{w}}{N + \alpha\overline{w}}\left\|\frac{g}{N} - \frac{g_w}{\overline{w}}\right\|^2 + d\log\frac{(N + \alpha\overline{w})}{N\alpha\overline{w}} + d\log N.$$

For $a, b > 0$ and $x \geq 0$, the function $ax - b\log x$ is convex in $x$ with minimum at $x^\star = b/a$, and so

$$\min_{\alpha \geq 0}(\dots) \geq d\log\left(N\left\|\frac{g}{N} - \frac{g_w}{\overline{w}}\right\|^2\right).$$

By assumption, $\|\frac{g}{N}\| = o_p(r)$ and $\|\frac{g_w}{\overline{w}}\| = o_p(r)$, and hence the $\alpha w$ branch has the asymptotic minimum:

$$\min_{\alpha \geq 0}\underline{\mathrm{KL}}(\alpha w) \geq O_p(1) + \Omega_p(1)\min\left\{-\log\pi(B^c), d\log\left(N\left\|\frac{g}{N} - \frac{g_w}{\overline{w}}\right\|^2\right)\right\}.$$

$\square$

*Proof of Theorem 3.5.* By Lemma 3.1,

$$\underline{\mathrm{KL}}(w) \geq -\log\min(1, J_B(w)) + \min(1, J_B(w)) - 1$$

$$\geq O_p(1) + \min\left\{G_B(w), -\log\sqrt{\pi(B^c)}\right\}$$

$$G_B(w) = -\log\int_B \pi_0\exp((1/2)(\ell + \ell_w)) + \frac{1}{2}\log\int\pi_0\exp(\ell) + \frac{1}{2}\log\int\pi_0\exp(\ell_w).$$

Note that $G_B$ in this proof is subtly different from the $G_B$ used in the proof of Theorem 3.3; the latter two integrals are over the whole space (directly from Lemma 3.1), rather than $B$. We shift the exponential arguments in $G_B(w)$ by $(1/2)(\ell(\theta_0) + \ell_w(\theta_0))$. We first provide lower bounds on two of the integral terms via Assumption 3.4:

$$\log\int\pi_0 e^{\ell-\ell(\theta_0)} \geq O_p(1) + \log\int e^{(g+g_0)^T(\theta-\theta_0) - \frac{(\sim 1)(N+1)L'^2}{2}\|\theta-\theta_0\|^2},$$

where $(\sim 1)$ denotes a quantity that converges in probability to 1, $g_0 = \nabla\log\pi_0(\theta_0)$, and $L'^2 = \frac{NL^2 + L_0^2}{N+1}$. Transforming variables via $x = L'(\theta - \theta_0)$,

$$\log\int\pi_0 e^{\ell-\ell(\theta_0)} \geq O_p(1) + \log\int e^{(g+g_0)^T x/L' - \frac{(\sim 1)(N+1)}{2}\|x\|^2}$$

$$= O_p(1) + \log\int e^{-\frac{(\sim 1)(N+1)}{2}\left\|x - \frac{g+g_0}{(N+1)L'}\right\|^2 + \frac{(\sim 1)(N+1)}{2}\left\|\frac{g+g_0}{(N+1)L'}\right\|^2}$$

$$= O_p(1) + \frac{(\sim 1)(N+1)}{2L'^2}\left\|\frac{g+g_0}{N+1}\right\|^2 - \frac{d}{2}\log(N+1)$$

$$\geq O_p(1) + \frac{(\sim 1)(N+1)}{2\max\{L^2, L_0^2\}}\left\|\frac{g+g_0}{N+1}\right\|^2 - \frac{d}{2}\log(N+1).$$

Let $L_w^2 = \frac{1}{\overline{w}}\sum_n w_n L_n^2$. Using the same technique, with $L_w'^2 = \frac{\overline{w}L_w^2 + L_0^2}{\overline{w}+1}$ and $x = L_w'(\theta - \theta_0)$,

$$\log\int\pi_0 e^{\ell_w - \ell_w(\theta_0)} \geq \log\int e^{(g_w+g_0)^T(\theta-\theta_0) - \frac{(\sim 1)(\overline{w}+1)}{2}L_w'^2\|\theta-\theta_0\|^2}$$

$$\geq O_p(1) + \frac{\overline{w}+1}{2L_w'^2}\left\|\frac{g_w+g_0}{\overline{w}+1}\right\|^2 + \log\int e^{-\frac{(\overline{w}+1)}{2}\left\|x - \frac{g_w+g_0}{(\overline{w}+1)L_w'}\right\|^2}$$

$$\geq O_p(1) + \frac{\overline{w}+1}{2L_w'^2}\left\|\frac{g_w+g_0}{\overline{w}+1}\right\|^2 + \log\int_{\|x - \frac{g_w+g_0}{(\overline{w}+1)L_w'}\| \leq (\overline{w}+1)^{-1/3}} e^{-\frac{\overline{w}+1}{2}\left\|x - \frac{g_w+g_0}{(\overline{w}+1)L_w'}\right\|^2}$$

$$= O_p(1) + \frac{\overline{w}+1}{2L_w'^2}\left\|\frac{g_w+g_0}{\overline{w}+1}\right\|^2 - \frac{d}{2}\log(\overline{w}+1)$$

$$\geq O_p(1) + \frac{\overline{w}+1}{2\max\{\beta L^2, L_0^2\}}\left\|\frac{g_w+g_0}{\overline{w}+1}\right\|^2 - \frac{d}{2}\log(\overline{w}+1).$$

For the upper bound on the first term, we use a local quadratic expansion around $\theta_0$, where $H_0 = -\nabla^2\log\pi_0(\theta_0)$,

$$\log\int_B \pi_0 e^{\frac{1}{2}(\ell-\ell(\theta_0)+\ell_w-\ell_w(\theta_0))} \leq O_p(1) + \log\int_B e^{\frac{1}{2}((g+g_w+2g_0)^T(\theta-\theta_0) - \frac{(\sim 1)(N+\overline{w}+2)}{4}(\theta-\theta_0)^T\left(\frac{(N+\alpha\overline{w})H+2H_0}{N+\overline{w}+2}\right)(\theta-\theta_0)}.$$

Because $H \succ 0$, we have $(N + \alpha\overline{w})H + 2H_0 \succ 0$ eventually; we can transform variables to $x = C^T(\theta - \theta_0)$, where $\frac{(N+\alpha\overline{w})H+2H_0}{N+\overline{w}+2} = CC^T$ is the Cholesky factorization, and subsequently complete the square. Note that

$$\sqrt{\min\{\min(\alpha, 1)\lambda_{\min}H, \lambda_{\min}H_0\}} \leq \lambda_{\min}C \leq \lambda_{\max}C \leq \sqrt{\max\{\max(\alpha, 1)\lambda_{\max}H, \lambda_{\max}H_0\}}$$

so

$$\log|C| = O_p(1) \qquad \lambda_{\min} C^{-1} H C^{-T} \geq \frac{\lambda_{\min} H}{\max\{\max(\alpha, 1)\lambda_{\max}H, \lambda_{\max}H_0\}} = \eta > 0,$$

and therefore

$$\log \int_B \pi_0 e^{\frac{1}{2}(\cdots)}$$

$$\leq O_p(1) + \frac{(\sim 1)(N + \overline{w} + 2)}{4}\left\|\frac{C^{-1}(g + g_w + 2g_0)}{N + \overline{w} + 2}\right\|^2 + \log \int_{\|x\|^2 \leq r^2 \eta^{-1}} e^{-\frac{(\sim 1)(N + \overline{w} + 2)}{4}\left\|x - \frac{(\sim 1)C^{-1}(g + g_w + 2g_0)}{N + \overline{w} + 2}\right\|^2}.$$

$$(11)$$

Suppose first that $\overline{w} + 1 \leq N/(4\|C^{-1}\|^2 \max\{\beta L^2, L_0^2\})$. In this case we bound the integral in Eq. (11) by integrating over the whole space:

$$\log \int_B \pi_0 e^{\frac{1}{2}(\cdots)} \leq O_p(1) + \frac{(\sim 1)\|C^{-1}\|^2(N + \overline{w} + 2)}{4}\left\|\frac{g + g_w + 2g_0}{N + \overline{w} + 2}\right\|^2 - \frac{d}{2}\log(N + \overline{w} + 2).$$

Combining this with the previous results yields

$$G_B(w) \geq O_p(1)$$

$$- \frac{(\sim 1)(N + \overline{w} + 2)}{4}\|C^{-1}\|^2\left\|\frac{g + g_w + 2g_0}{N + \overline{w} + 2}\right\|^2$$

$$+ \frac{d}{4}\log\frac{(N + \overline{w} + 2)^2}{(N + 1)(\overline{w} + 1)} + \frac{\overline{w} + 1}{4\max\{\beta L^2, L_0^2\}}\left\|\frac{g_w + g_0}{\overline{w} + 1}\right\|^2 + \frac{(N + 1)}{4\max\{L^2, L_0^2\}}\left\|\frac{g + g_0}{N + 1}\right\|^2$$

$$\geq O_p(1) + \frac{d}{4}\log\frac{(N + \overline{w} + 2)^2}{(N + 1)(\overline{w} + 1)} + \frac{\overline{w} + 1}{4}\left\|\frac{g_w + g_0}{\overline{w} + 1}\right\|^2\left(\frac{1}{\max\{\beta L^2, L_0^2\}} - \frac{2\|C^{-1}\|^2(\overline{w} + 1)}{N + \overline{w} + 2}\right)$$

$$\geq O_p(1) + \frac{d}{4}\log\frac{(N + \overline{w} + 2)^2}{(N + 1)(\overline{w} + 1)} + \frac{\overline{w} + 1}{8\max\{\beta L^2, L_0^2\}}\left\|\frac{g_w + g_0}{\overline{w} + 1}\right\|^2.$$

Bounding the last term below by 0 and minimizing over $w$ such that $\overline{w} \leq \sqrt{N}$ yields

$$G_B(w) \geq O_p(1) + \frac{d}{4}\log\sqrt{N} = O_p(1) + \frac{d}{8}\log N.$$

Bounding $(N + \overline{w} + 2)/(N + 1) \geq 1$ and minimizing over $w$ such that $\overline{w} \geq \sqrt{N}$ yields

$$G_B(w) \geq O_p(1) + \frac{d}{4}\log N - \frac{d}{4}\log(\overline{w} + 1) + \frac{\overline{w} + 1}{8\max\{\beta L^2, L_0^2\}}\left\|\frac{g_w + g_0}{\overline{w} + 1}\right\|^2$$

$$\geq O_p(1) + \frac{d}{4}\log N\left\|\frac{g_w + g_0}{\overline{w} + 1}\right\|^2$$

$$= O_p(1) + \frac{d}{4}\log N\left\|\frac{g_w}{\overline{w}}\right\|^2,$$

where the second line follows because for $a, b > 0$ and $x \geq 0$, the function $ax - b\log x$ is convex in $x$ with minimum at $x^\star = b/a$. Therefore for $\overline{w} + 1 \leq N/(\ldots)$,

$$\underline{\text{KL}}(w) \geq O_p(1) + \Omega_p(1)d\log\left(N\min\left\{\left\|\frac{g_w}{\overline{w}}\right\|^2, 1\right\}\right).$$

Next suppose $\overline{w} + 1 \geq N/(4\|C^{-1}\|^2\max\{\beta L^2, L_0^2\})$. A second upper bound on Eq. (11) can be obtained by taking the maximum of the integrand over the integration region $\|x\|^2 \leq r^2$. Note that since $\|g_w/\overline{w}\| = \omega_p(r)$, $\overline{w} = \Omega_p(N)$, $g/N = O_p(N^{-1/2})$, and $Nr^2 = \omega_p(1)$, we have that $\|(g + g_w + 2g_0)/(N + \overline{w} + 2)\| = \omega_p(r)$, and so

$$\log \int_B \pi_0 e^{\frac{1}{2}(\cdots)}$$

$$\leq O_p(1) + \frac{(\sim 1)(N + \overline{w} + 2)}{4}\left\|\frac{C^{-1}(g + g_w + 2g_0)}{N + \overline{w} + 2}\right\|^2 - \frac{(\sim 1)(N + \overline{w} + 2)}{4}\left(\left\|\frac{C^{-1}(g + g_w + 2g_0)}{N + \overline{w} + 2}\right\| - r\right)^2 + \frac{d}{2}\log r^2$$

$$= O_p(1) - \frac{(\sim 1)(N + \overline{w} + 2)}{4}r^2 + \frac{(\sim 1)(N + \overline{w} + 2)r}{2}\left\|\frac{C^{-1}(g + g_w + 2g_0)}{N + \overline{w} + 2}\right\| + \frac{d}{2}\log r^2.$$

So therefore combining this result with the previous bounds and minimizing over $\overline{w}$ yields

$$G_B(w) \geq O_p(1) + \frac{(\sim 1)(N + \overline{w} + 2)}{4}r^2 - \frac{(\sim 1)(N + \overline{w} + 2)r}{2}\left\|\frac{C^{-1}(g + g_w + 2g_0)}{N + \overline{w} + 2}\right\|$$

$$-\frac{d}{4}\log((N+1)(\overline{w}+1)r^4)+\frac{\overline{w}+1}{4\max\{\beta L^2,L_0^2\}}\left\|\frac{g_w+g_0}{\overline{w}+1}\right\|^2+\frac{(N+1)}{4\max\{L^2,L_0^2\}}\left\|\frac{g+g_0}{N+1}\right\|^2$$

$$\geq O_p(1)-\frac{d}{4}\log(Nr^2)+\frac{(\sim 1)N}{4}\left(\left\|\frac{g}{N}\right\|-r\right)^2-\frac{d}{4}\log(\overline{w}r^2)+\frac{(\sim 1)\overline{w}}{4}\left(\left\|\frac{g_w}{\overline{w}}\right\|-r\right)^2$$

$$\geq O_p(1)-\frac{d}{4}\log(Nr^2)+\frac{(\sim 1)}{4}Nr^2-\frac{d}{4}\log(r^2)+\frac{d}{4}\log\left\|\frac{g_w}{\overline{w}}\right\|^2$$

$$\geq O_p(1)+\frac{d}{4}\log N\left\|\frac{g_w}{\overline{w}}\right\|^2.$$

Combining with the earlier bound and noting that $N\min\{\|g_w/\overline{w}\|,1\}=\omega_p(1)$ yields the final result. □

*Proof of Corollary 3.6.* The proof follows directly from Theorems 3.3 and 3.5 by the data processing inequality applied to $\underline{\mathrm{KL}}(w)$. □

*Proof of Theorem 5.3.* By Lemma 5.1,

$$\overline{\mathrm{KL}}(w)\leq\inf_{\lambda>0}\frac{1}{\lambda}\log\int\pi\exp\big((1+\lambda)(\bar{\ell}_w-\bar{\ell})\big)$$

$$=\inf_{\lambda>0}\frac{1}{\lambda}\log\int\pi\exp\big(\bar{\ell}_{(1+\lambda)(w-1)}\big).$$

Since $(\ell_n)_{n=1}^N$ are $(f,A)$-subexponential, if

$$(1+\lambda)^2(w-1)^T A(w-1)\leq 1,$$

then

$$\int\pi\exp\big(\bar{\ell}_{(1+\lambda)(w-1)}\big)\leq\exp\Big(f((1+\lambda)^2(w-1)^T A(w-1))\Big).$$

By assumption, the condition holds when $\lambda=1$; the result follows. □

*Proof of Proposition 5.4.* Let $C(w)=\log\int\pi\exp(\bar{\ell}_w)$. By the finiteness condition, [53, Theorem 2.4] asserts that $C(w)$ is continuous, and has derivatives of all orders that can be obtained by passing differentiation through the integral within the set $\|w\|_2<\alpha$. Let $U=\mathrm{Cov}_\pi((\ell_n)_{n=1}^N)$, and $\mathcal{S}=\mathrm{span}\{w\in\mathbb{R}^N:w^T(\bar{\ell}_n)_{n=1}^N=0\ \pi\text{-a.s.}\}$. Note that $\mathcal{S}=\ker U$: since $w^T U w=\mathrm{Var}_\pi(w^T(\ell_n)_{n=1}^N)$, $w^T U w=0$ if and only if $w^T(\bar{\ell}_n)_{n=1}^N=0$ $\pi$-a.s.; and since $U$ is symmetric positive semidefinite, $w^T U w=0$ if and only if $w\in\ker U$. Therefore $C(w)$ is continuous, has derivatives of all orders, and derivatives can be passed through the integral within the set $\{w\in\mathbb{R}^N:w=v+u,\|v\|_2<\alpha/2,u\in\ker U\}$. For a vector $w=v+u$, $v\perp\ker U$, $u\in\ker U$, and minimum positive eigenvalue $\lambda_+$ of $U$,

$$w^T U w\leq\frac{\alpha^2\lambda_+}{4}\implies v^T U v\leq\frac{\alpha^2\lambda_+}{4}\implies\|v\|_2\leq\frac{\alpha}{2},$$

and so $C(w)$ is continuous, has derivatives of all orders, and derivatives can be passed through the integral within the set $\{w\in\mathbb{R}^N:w^T U w\leq\frac{\alpha^2\lambda_+}{4}\}$. By Taylor's theorem, for any $w$ in this set, there exists a distribution $\nu_w$ with density proportional to $\pi\exp(\bar{\ell}_{w'})$ for some $w'$ on the segment from the origin to $w$ such that

$$C(w)=\log\int\pi\exp(\bar{\ell}_w)=\frac{1}{2}w^T U w+\frac{1}{6}\mathbb{E}_{\nu_w}\Big[(w^T(\bar{\ell}_n)_{n=1}^N)^3\Big].$$

By definition of $\nu_w$, $w\in\ker U$ implies that $w^T(\bar{\ell}_n)_{n=1}^N=0$ $\nu_w$-a.s. and hence $\frac{1}{6}\mathbb{E}_{\nu_w}\big[(w^T(\bar{\ell}_n)_{n=1}^N)^3\big]=0$. Therefore, for $w^T U w\leq\frac{\alpha^2\lambda_+}{4}$,

$$C(w)\leq\frac{1}{2}w^T U w\left(1+\max_{\substack{w^T U w\leq\frac{\alpha^2\lambda_+}{4}\\ w\perp\ker U}}\frac{1}{6}\frac{\mathbb{E}_{\nu_w}\big[(w^T(\bar{\ell}_n)_{n=1}^N)^3\big]}{w^T U w}\right)$$

$$\leq\frac{1}{2}w^T U w\left(1+\max_{\|w\|_2\leq\frac{\alpha}{2}}\frac{1}{6}\frac{\|w\|_2\big\|\mathbb{E}_{\nu_w}\big[(\bar{\ell}_n)_{n=1}^N\otimes(\bar{\ell}_n)_{n=1}^N\otimes(\bar{\ell}_n)_{n=1}^N\big]\big\|_2}{\lambda_+}\right)$$

$$\leq\frac{1}{2}w^T U w\left(1+\frac{\alpha}{12\lambda_+}\max_{\|w\|_2\leq\frac{\alpha}{2}}\Big\|\mathbb{E}_{\nu_w}\big[(\bar{\ell}_n)_{n=1}^N\otimes(\bar{\ell}_n)_{n=1}^N\otimes(\bar{\ell}_n)_{n=1}^N\big]\Big\|\right),$$

where $\otimes$ denotes outer products to form a tensor. By continuity of derivatives of all orders within the neighbourhood $\|w\|_2<\alpha$, the result follows by selecting a sufficiently small $\alpha$. □

**Proposition A.1.** *Suppose there exist $c \in \mathbb{R}$, $\alpha, \delta > 0$, and $0 < \epsilon < 1$ such that $\ell \le c$ and for all coreset weights $w$ satisfying $\alpha w^T \operatorname{Cov}_\pi((\ell_n)_{n=1}^N)w \le 1$, $|\bar{\ell}_w| \le \epsilon|\ell - c| + \delta$. Then the potentials $(\ell_n)_{n=1}^N$ are $(h, \alpha \operatorname{Cov}_\pi((\ell_n)_{n=1}^N))$-subexponential with $h(x) = \frac{1}{2}x + \frac{e^{\delta+c\epsilon}}{\int \pi_0 e^{\epsilon\ell}} x^{1-\epsilon}$.*

*Proof of Proposition A.1.* Let $\ell' = \ell - c$. Since $\ell' \le 0$ and $|\bar{\ell}_w| \le \epsilon|\ell'| + \delta$ for some $\epsilon < 1$, $\delta > 0$,

$$
\int \pi \exp(\bar{\ell}_w) = 1 + \frac{1}{2}\int \pi(\bar{\ell}_w)^2 + \int \pi \sum_{k=3}^\infty \frac{1}{k!}(\bar{\ell}_w)^{k-2(1-\epsilon)}(\bar{\ell}_w)^{2(1-\epsilon)}
$$

$$
\le 1 + \frac{1}{2}\int \pi(\bar{\ell}_w)^2 + \int \pi \sum_{k=3}^\infty \frac{1}{k!}(\epsilon|\ell'| + \delta)^{k-2(1-\epsilon)}|\bar{\ell}_w|^{2(1-\epsilon)}
$$

$$
= 1 + \frac{1}{2}\int \pi(\bar{\ell}_w)^2 + \int \pi \left( \frac{e^{\epsilon|\ell'|+\delta} - 1 - (\epsilon|\ell'| + \delta) - \frac{1}{2}(\epsilon|\ell'| + \delta)^2}{(\epsilon|\ell'| + \delta)^{2(1-\epsilon)}} \right)|\bar{\ell}_w|^{2(1-\epsilon)}
$$

$$
\le 1 + \frac{1}{2}\int \pi(\bar{\ell}_w)^2 + \int \pi e^{\epsilon|\ell'|+\delta}|\bar{\ell}_w|^{2(1-\epsilon)}
$$

$$
= 1 + \frac{1}{2}\int \pi(\bar{\ell}_w)^2 + \frac{\int \pi_0 e^{(1-\epsilon)\ell'+\delta}|\bar{\ell}_w|^{2(1-\epsilon)}}{\int \pi_0 e^{\ell'}}
$$

$$
\le 1 + \frac{1}{2}\int \pi(\bar{\ell}_w)^2 + e^\delta \frac{\left( \int \pi_0 e^{\ell'}|\bar{\ell}_w|^2 \right)^{1-\epsilon}}{\int \pi_0 e^{\ell'}}
$$

$$
= 1 + \frac{1}{2}\int \pi(\bar{\ell}_w)^2 + e^\delta \left( \int \pi_0 e^{\ell'} \right)^{-\epsilon} \left( \int \pi(\bar{\ell}_w)^2 \right)^{1-\epsilon}
$$

$$
= 1 + \frac{1}{2}\int \pi(\bar{\ell}_w)^2 + e^{\delta+c\epsilon} \left( \int \pi_0 e^{\ell} \right)^{-\epsilon} \left( \int \pi(\bar{\ell}_w)^2 \right)^{1-\epsilon}
$$

$$
\le \exp\left( h(w^T \operatorname{Cov}_\pi(\ell)w) \right),
$$

where $h(x) = \frac{1}{2}x + \frac{e^{\delta+c\epsilon}}{\int \pi_0 e^{\epsilon\ell}}x^{1-\epsilon}$. $\qquad\square$

**Proposition A.2.** *Suppose $\Theta = \mathbb{R}^d$, $\bar{\ell}$ is $G$-strongly concave, and there exists $L < G$, $\alpha > 0$, and $\theta_0 \in \Theta$ such that for all coreset weights $w$ satisfying $\alpha w^T \operatorname{Cov}_\pi((\ell_n)_{n=1}^N)w \le 1$, $\bar{\ell}_w$ is $L$-Lipschitz smooth, and both $\|\nabla \ell_w(\theta_0)\|$ and $\bar{\ell}_w(\theta_0)$ are bounded. Then for any $(L/G) < \epsilon < 1$, there exists $c \in \mathbb{R}$, $\delta > 0$ such that the potentials $(\ell_n)_{n=1}^N$ are $(h, \alpha \operatorname{Cov}_\pi((\ell_n)_{n=1}^N))$-subexponential with the same $h$ as in Proposition A.1.*

*Proof of Proposition A.2.* Since $\bar{\ell}$ is $G$-strongly concave and $\bar{\ell}_w$ is $L$-Lipschitz smooth, we can write

$$
\ell(\theta) \le \ell(\theta_0) + \nabla \ell(\theta_0)^T(\theta - \theta_0) - \frac{G}{2}\|\theta - \theta_0\|^2
$$

$$
= \ell(\theta_0) + \frac{G}{2}\left\|G^{-1}\nabla\ell(\theta_0)\right\|^2 - \frac{G}{2}\left\|\theta - \theta_0 - G^{-1}\nabla\ell(\theta_0)\right\|^2
$$

$$
|\bar{\ell}_w(\theta)| \le |\bar{\ell}_w(\theta_0) + \nabla\ell_w(\theta_0)^T(\theta - \theta_0)| + \frac{L}{2}\|\theta - \theta_0\|^2.
$$

So setting $c = \ell(\theta_0) + \frac{G}{2}\left\|G^{-1}\nabla\ell(\theta_0)\right\|^2$ implies $\ell - c$ is a nonpositive function as required. Then

$$
|\bar{\ell}_w(\theta)| - \epsilon|\ell(\theta) - c| \le |\bar{\ell}_w(\theta_0)| + \frac{\epsilon}{2G}\|\nabla\ell(\theta_0)\|^2 + (\|\nabla\ell_w(\theta_0)\| + \epsilon\|\nabla\ell(\theta_0)\|)\|\theta - \theta_0\| + \frac{L - \epsilon G}{2}\|\theta - \theta_0\|^2.
$$

For $0 < a < G - L$, setting $\epsilon = \frac{L+a}{G}$ and then maximizing over $\|\theta - \theta_0\|$ yields

$$
|\bar{\ell}_w(\theta)| - \epsilon|\ell(\theta) - c| \le |\bar{\ell}_w(\theta_0)| + \frac{\epsilon}{2G}\|\nabla\ell(\theta_0)\|^2 + (\|\nabla\ell_w(\theta_0)\| + \epsilon\|\nabla\ell(\theta_0)\|)\|\theta - \theta_0\| - \frac{a}{2}\|\theta - \theta_0\|^2
$$

$$
\le |\bar{\ell}_w(\theta_0)| + \frac{\epsilon}{2G}\|\nabla\ell(\theta_0)\|^2 + \frac{(\|\nabla\ell_w(\theta_0)\| + \epsilon\|\nabla\ell(\theta_0)\|)^2}{2a}.
$$

By the boundedness of $\bar{\ell}_w(\theta_0)$ and $\nabla\ell_w(\theta_0)$, maximizing over $w$ yields a value of $\delta < \infty$. $\qquad\square$

**Lemma A.3.** *Let $X_1, X_2, \ldots$ be i.i.d. random variables in $\mathbb{R}$ with $\mathbb{E}X_n = 0$, and define the resampled sum*

$$
S_N = \sum_{n=1}^N \frac{M_n}{Mp_n}X_n
$$

where $(M_1, \ldots, M_N) \sim \text{Multi}(M, (p_1, \ldots, p_N))$, *with strictly positive resampling probabilities* $p_1, \ldots, p_N$ *that may depend on* $X_1, \ldots, X_N$ *and* $N$. *If there exists a* $\delta > 0$ *such that as* $N \to \infty$,

$$\frac{1}{N} \sum_n \frac{|X_n|^{2+\delta}}{(Np_n)^{1+\delta}} = O_p(1), \quad \frac{1}{N} \sum_n \frac{X_n^2}{Np_n} = \Omega_p(1), \quad and \quad M \to \infty,$$

*then*

$$\sqrt{M} \frac{\frac{1}{N} S_N - \frac{1}{N} \sum_n X_n}{\sqrt{\frac{1}{N} \sum_n \frac{X_n^2}{Np_n}}} \xrightarrow{d} \mathcal{N}(0, 1).$$

*Proof.* We can rewrite

$$S_N = \frac{1}{M} \sum_{m=1}^{M} \frac{X_{I_m}}{p_{I_m}}$$

where $I_m \overset{\text{iid}}{\sim} \text{Categorical}(p_1, \ldots, p_N)$. Consider $S_N + B_N$ where $B_N$ is independent of $S_N$, $B_N = \pm 1$ with probability $\frac{1}{2(NM)^{1+\delta}}$, and $B_N = 0$ otherwise. So if we set $\mathcal{A}_N = \sigma(X_1, \ldots, X_N)$, [54, Cor. 3] asserts that

$$\frac{S_N + B_N - \mathbb{E}[S_N | \mathcal{A}_N]}{\sqrt{(NM)^{-(1+\delta)} + \text{Var}[S_N | \mathcal{A}_N]}} \xrightarrow{d} \mathcal{N}(0, 1) \qquad N \to \infty.$$

as long as for all $N$ large enough,

$$\text{Var}\left[\frac{1}{M} \frac{X_{I_m}}{p_{I_m}} \Big| \mathcal{A}_N\right] < \infty \quad \text{a.s.},$$

and as $N \to \infty$,

$$\frac{(NM)^{-(1+\delta)} + \sum_{m=1}^{M} \mathbb{E}\left[\left|\frac{1}{M} \frac{X_{I_m}}{p_{I_m}} - \mathbb{E}\left[\frac{1}{M} \frac{X_{I_m}}{p_{I_m}} \Big| \mathcal{A}_N\right]\right|^{2+\delta} \Big| \mathcal{A}_N\right]}{\left((NM)^{-(1+\delta)} + \text{Var}[S_N | \mathcal{A}_N]\right)^{(2+\delta)/2}} \xrightarrow{p} 0.$$

Note that the conditional mean and variance have the form

$$\mathbb{E}[S_N | \mathcal{A}_N] = \mathbb{E}\left[\frac{X_{I_m}}{p_{I_m}} \Big| \mathcal{A}_N\right] = \sum_n X_n$$

$$\text{Var}[S_N | \mathcal{A}_N] = \frac{1}{M} \text{Var}\left[\frac{X_{I_m}}{p_{I_m}} \Big| \mathcal{A}_N\right] = \frac{1}{M} \sum_n p_n \left(\frac{X_n}{p_n} - \sum_n X_n\right)^2,$$

which implies that $\text{Var}\left[\frac{1}{M} \frac{X_{I_m}}{p_{I_m}} | \mathcal{A}_N\right] < \infty$ a.s., since $p_1, \ldots, p_N$ are strictly nonnegative and $\mathbb{E}X_n = 0$ implies $X_n$ is finite almost surely. Next, note that

$$\frac{(NM)^{-(1+\delta)} + \sum_{m=1}^{M} \mathbb{E}\left[\left|\frac{1}{M} \frac{X_{I_m}}{p_{I_m}} - \mathbb{E}\left[\frac{1}{M} \frac{X_{I_m}}{p_{I_m}} \Big| \mathcal{A}_N\right]\right|^{2+\delta} \Big| \mathcal{A}_N\right]}{\left((NM)^{-(1+\delta)} + \text{Var}[S_N | \mathcal{A}_N]\right)^{(2+\delta)/2}}$$

$$\leq \frac{(NM)^{-(1+\delta)} + 2^{2+\delta} \sum_{m=1}^{M} \left(\mathbb{E}\left[\left|\frac{1}{M} \frac{X_{I_m}}{p_{I_m}}\right|^{2+\delta} \Big| \mathcal{A}_N\right] + \left|\frac{1}{M} \sum_n X_n\right|^{2+\delta}\right)}{\left((NM)^{-(1+\delta)} + \text{Var}[S_N | \mathcal{A}_N]\right)^{(2+\delta)/2}}$$

$$= M^{-\delta/2} \frac{N^{-(3+2\delta)} + 2^{2+\delta}\left(\frac{1}{N} \sum_n \frac{|X_n|^{2+\delta}}{(Np_n)^{1+\delta}} + \left|\frac{1}{N} \sum_n X_n\right|^{2+\delta}\right)}{\left(M^{-\delta} N^{-(3+\delta)} + \frac{1}{N} \sum_n \frac{X_n^2}{Np_n} - \left(\frac{1}{N} \sum_n X_n\right)^2\right)^{(2+\delta)/2}}.$$

The above expression converges in probability to 0 by the technical assumptions in the statement of the result as well as the fact that $\frac{1}{N} \sum_n X_n \overset{a.s.}{\to} 0$ by the law of large numbers. Once again by the technical assumptions, $\text{Var}[S_N | \mathcal{A}_N] = \Omega_p(N^2/M)$, so

$$\frac{\text{Var}[S_N | \mathcal{A}_N]}{(NM)^{-(1+\delta)} + \text{Var}[S_N | \mathcal{A}_N]} \xrightarrow{p} 1$$

$$\frac{B_N}{(NM)^{-(1+\delta)} + \text{Var}[S_N | \mathcal{A}_N]} \xrightarrow{p} 0,$$

and hence by Slutsky's theorem,

$$\frac{S_N - \mathbb{E}[S_N | \mathcal{A}_N]}{\sqrt{\text{Var}[S_N | \mathcal{A}_N]}} \xrightarrow{d} \mathcal{N}(0, 1) \qquad N \to \infty.$$

Using Slutsky's theorem again with $\frac{1}{N} \sum_n X_n \xrightarrow{p} 0$ and rearranging yields the final result. $\qquad \square$

**Lemma A.4.** *Suppose coreset weights are generated using the importance weighted construction in Algorithm 1. Let $g = \nabla \ell(\eta_0)$, $g_w = \nabla \ell_w(\eta_0)$, and $H = -\mathbb{E}\left[\nabla^2 \ell_n(\eta_0)\right]$. If conditions A(1-3) and (A6) in Section 4 hold, $M = o(N)$, and $M = \omega(1)$, then*

$$\left\|\frac{g}{N}\right\|_2 = \Theta_p\left(N^{-1/2}\right), \qquad \left\|\frac{g_w}{\overline{w}}\right\|_2 = \Theta_p\left(M^{-1/2}\right), \qquad \frac{\overline{w}}{N} \xrightarrow{p} 1,$$

*and*

$$\sup_{\|\eta - \eta_0\|_2 \leq r} \left\|-\frac{1}{N}\nabla^2 \ell(\eta) - H\right\|_2 \xrightarrow{p} 0, \qquad \sup_{\|\eta - \eta_0\|_2 \leq r} \left\|-\frac{1}{\overline{w}}\nabla^2 \ell_w(\eta) - H\right\|_2 \xrightarrow{p} 0.$$

*Proof.* First, since $\overline{w} = \sum_n \frac{M_n}{M p_n}$, $\mathbb{E}\overline{w} = N$, and

$$\mathbb{E}\left[(\overline{w} - N)^2\right] = \frac{N^2}{M^2}\mathbb{E}\left[\left(\sum_n M_n\left((Np_n)^{-1} - 1\right)\right)^2\right]$$

$$= \frac{N^2}{M^2}\left(\sum_n ((Np_n)^{-1} - 1)^2 \mathbb{E}M_n^2 + \sum_{n \neq n'} ((Np_n)^{-1} - 1)((Np_{n'})^{-1} - 1)\mathbb{E}[M_n M_{n'}]\right)$$

$$= \frac{N^2}{M}\left(\sum_n ((Np_n)^{-1} - 1)^2 p_n - \left(\sum_{n'}(1/N - p_n)\right)^2\right)$$

$$= \frac{1}{M}\left(\sum_n p_n(p_n^{-1} - N)^2\right)$$

$$\leq \frac{1}{M}\left(\max_n (p_n^{-1} - N)^2\right)$$

$$\leq \frac{N^2}{M}O(1),$$

where the last line follows by assumption A6. Therefore by Chebyshev's inequality and $M \to \infty$, $\overline{w}/N \xrightarrow{p} 1$. Since the data are i.i.d., by conditions A1 and A2, the central limit theorem holds for the sum of $\nabla \ell_n(\eta_0)$ such that $g/\sqrt{N}$ converges in distribution to a normal, and hence $\left\|\frac{g}{N}\right\| = \Theta_p(N^{-1/2})$. By conditions A1, A2, and A6, Lemma A.3 holds such that for any $t \in \mathbb{R}^d$,

$$\sqrt{M}\,\frac{\frac{1}{N}t^T g_w - \frac{1}{N}t^T g}{\sqrt{\frac{1}{N}\sum_n \frac{(t^T \nabla \ell_n(\eta_0))^2}{Np_n}}} = \Theta_p(1).$$

Since condition A6 asserts that $C > Np_n \geq c > 0$, the law of large numbers, condition A1, and $M/N \to 0$ imply that

$$\frac{\sqrt{M}}{N}t^T g_w = \Theta_p(1).$$

Summing over a basis of vectors $t_1, \ldots, t_d$ shows that

$$\frac{\sqrt{M}}{N}\|g_w\|_2 = \Theta(1)\sqrt{M}\left\|\frac{g_w}{\overline{w}}\right\|_2 \Theta_p(1).$$

This completes the first three results. Next, by condition A3, for sufficiently large $N$ such that the neighbourhood contains the ball of radius $r$ around $\eta_0$,

$$\sup_{\|\eta - \eta_0\|_2 \leq r} \left\|\frac{1}{N}\nabla^2 \ell(\eta) - \frac{1}{N}\nabla^2 \ell(\eta_0)\right\|_2 \leq r\frac{1}{N}\sum_n R(X_n)$$

$$\sup_{\|\eta - \eta_0\|_2 \leq r} \left\|\frac{1}{N}\nabla^2 \ell_w(\eta) - \frac{1}{N}\nabla^2 \ell_w(\eta_0)\right\|_2 \leq r\frac{1}{N}\sum_n w_n R(X_n),$$

and

$$\mathbb{E}\left[r\frac{1}{N}\sum_n R(X_n)\right] = \mathbb{E}\left[r\frac{1}{N}\sum_n w_n R(X_n)\right] = r\mathbb{E}\left[R(X)\right] \to 0,$$

so that we have that both

$$\sup_{\|\eta - \eta_0\|_2 \leq r} \left\|\frac{1}{N}\nabla^2 \ell(\eta) - \frac{1}{N}\nabla^2 \ell(\eta_0)\right\|_2 \xrightarrow{p} 0 \quad \text{and} \quad \sup_{\|\eta - \eta_0\|_2 \leq r} \left\|\frac{1}{N}\nabla^2 \ell_w(\eta) - \frac{1}{N}\nabla^2 \ell_w(\eta_0)\right\|_2 \xrightarrow{p} 0$$

by Markov's inequality. Finally, by the bounded variance in A2, sampling probability bounds in A6, and $M \to \infty$, the variances of $\frac{1}{N}\nabla^2 \ell_w(\eta_0)$ and $\frac{1}{N}\nabla^2 \ell(\eta_0)$ both converge to 0 as $N \to \infty$, and since both of these quantities are unbiased estimates of $\mathbb{E}[\nabla^2 \ell_n(\eta_0)]$, Chebyshev's inequality yields the desired convergence in probability. $\qquad\square$

**Lemma A.5.** *Suppose* $(X_n)_{n=1}^N$ *are* $N$ *i.i.d. random vectors in* $\mathbb{R}^d$. *Fix* $M \in \mathbb{N}$, $M < d$ *and define* $X = \begin{bmatrix} X_1 & X_2 & \dots & X_M \end{bmatrix} \in \mathbb{R}^{d \times M}$. *If there exists* $\delta > 0$ *such that*

$$\mathbb{E}\left[ (1^T (X^T X)^{-1} 1)^{M+\delta} \right] < \infty,$$

*where* 1 *denotes a vector of all 1 entries, then as* $N \to \infty$,

$$\left( \min_{w \in \mathbb{R}_+^N} \left\| \frac{\sum_{n=1}^N w_n X_n}{\sum_{n=1}^N w_n} \right\|^2 \quad s.t. \sum_n \mathbb{1}[w_n > 0] < M. \right) = \omega_p\left( N^{-\frac{M+\delta/2}{M+\delta}} \right).$$

*Proof.* For any $\epsilon > 0$, by the union bound over subsets of $[N]$ of size $M$,

$$\mathbb{P}\left( \min_{w \in \mathbb{R}_+^N} \dots \le \epsilon \right) \le \binom{N}{M} \mathbb{P}\left( \min_{w \in \mathbb{R}^M} \frac{w^T X^T X w}{w^T 1 1^T w} \le \epsilon \right)$$

$$\le \binom{N}{M} \mathbb{P}\left( \max_\lambda \min_{w \in \mathbb{R}^M} w^T X^T X w - \lambda(1^T w - 1) \le \epsilon \right)$$

$$= \binom{N}{M} \mathbb{P}\left( \max_\lambda \lambda - \frac{\lambda^2}{4} 1^T (X^T X)^{-1} 1 \le \epsilon \right)$$

$$= \binom{N}{M} \mathbb{P}\left( 1^T (X^T X)^{-1} 1 \ge \epsilon^{-1} \right).$$

By Markov's inequality and $\binom{N}{M} \le (eN/M)^M$,

$$\mathbb{P}\left( \min_{w \in \mathbb{R}_+^N} \dots \le \epsilon \right) \le \left( \frac{eN}{M} \right)^M \epsilon^{M+\delta} \mathbb{E}\left[ (1^T(X^T X)^{-1} 1)^{M+\delta} \right]$$

$$= \left( \frac{eN\epsilon^{\frac{M+\delta}{M}}}{M} \right)^M \mathbb{E}\left[ (1^T(X^T X)^{-1} 1)^{M+\delta} \right].$$

Setting $\epsilon = N^{-\frac{M+\delta/2}{M+\delta}}$ yields

$$\mathbb{P}\left( \min_{w \in \mathbb{R}_+^N} \dots \le N^{-\frac{M+\delta/2}{M+\delta}} \right) \le \left( \frac{eN^{-\frac{\delta}{2M}}}{M} \right)^M \mathbb{E}\left[ (1^T(X^T X)^{-1} 1)^{M+\delta} \right].$$

The right-hand side converges to 0 as $N \to \infty$, yielding the stated result. $\qquad\square$

*Proof of Corollary 4.1 and Corollary 4.2.* Set $r = (\log M)^{-1/2}$. Then since $M = o(N)$, $M = \omega(1)$, and assumptions (A1-3) and (A6) hold, Lemma A.4 holds. Note that $\|g_w/\overline{w}\| = \Theta_p(M^{-1/2}) = o_p(r)$, $\eta\pi_0$ is positive at $\eta_0$ and twice differentiable by (A4), and $Nr^2 = N/\log M = \omega(1)$ since $M = o(N)$. Thus the conditions of Theorem 3.3 are verified. Substitution into the right term in the minimum of Theorem 3.3 yields the stated lower bound of $\Omega_p(N/M)$. For the left term in Theorem 3.3, define $B = \{(\eta - \eta_0)^T H(\eta - \eta_0) \le r^2\}$. Then since $H \succ 0$, $r \to 0$, and $r^2 = 1/\log M = \omega(\log N/N)$, (A5) guarantees that $-\log(\eta\pi)(B^c) = \Omega_p(Nr^2) = \Omega_p(N/\log M)$. Therefore the minimum is $\Omega_p(N/M)$, and we complete the proof by transferring from $\underline{KL}(w)$ on the $\eta$-pushforward model to $\underline{KL}(w)$ on the original model using Corollary 3.6. $\qquad\square$

*Proof of Corollary 4.3.* Fix the $\delta > 0$ guaranteed by (A8), and set $r = N^{-\frac{M+3\delta/4}{2(M+\delta)}}$. Note that $Nr^2 = N^{\frac{\delta/4}{M+\delta}} = \omega(1)$, $\eta\pi_0$ is positive at $\eta_0$ and twice differentiable by (A4), and by (A1-3) the results pertaining to $\left\| \frac{g}{N} \right\|_2$ and $\sup_{\|\eta - \eta_0\|_2 \le r} \left\| -\frac{1}{N}\nabla^2 \ell(\eta) - H \right\|_2$ in Lemma A.4 hold; thus Assumption 3.2 holds. By (A7), Assumption 3.4 holds as well as the conditions on $\frac{1}{\overline{w}}\nabla^2 \ell_w(\theta)$ and $\frac{1}{\overline{w}}\sum_n w_n L_n^2$ in Theorem 3.5. Finally by (A8), Lemma A.5 holds such that

$$\left\| \frac{g_w}{\overline{w}} \right\|^2 = \omega_p\left( N^{-\frac{M+\delta/2}{M+\delta}} \right),$$

and hence $\left\|\frac{g_w}{\bar{w}}\right\| = \omega_p(r)$. Therefore all conditions of Theorem 3.5 hold. For the left term in the minimum in Theorem 3.5, define $B = \{(\eta - \eta_0)^T H(\eta - \eta_0) \leq r^2\}$. Then since $H \succ 0$, $r \to 0$, and $r^2 = N^{-\frac{M+3\delta/4}{M+\delta}} = \omega(\log N/N)$, (A5) guarantees that $-\log(\eta\pi)(B^c) = \Omega_p(Nr^2) = \Omega_p\left(N^{\frac{\delta/4}{M+\delta}}\right)$. For the right term,

$$\log\left(N\left\|\frac{g_w}{\bar{w}}\right\|^2\right) = \Omega_p\left(\log N^{1-\frac{M+\delta/2}{M+\delta}}\right) = \Omega_p(\log N).$$

The minimum of these two is from the right term, so

$$\underline{\text{KL}}(w) = \Omega_p(\log N).$$

We complete the proof by transferring from $\underline{\text{KL}}(w)$ on the $\eta$-pushforward model to $\underline{\text{KL}}(w)$ on the original model using Corollary 3.6. $\qquad \square$

**Proposition A.6.** *The models specified in Eqs.* (2) *and* (3) *satisfy assumptions (A1-5).*

*Proof.* The exact same technique applies to both models, so here we will just demonstrate it for the Cauchy model. In the Cauchy model, $\theta \in \mathbb{R}$, $\eta : \mathbb{R} \to \mathbb{R}_+$, $\eta(\theta) = \theta^2$, and

$$\ell_n(\eta) = -\log\pi - \log((Z_n - \eta)^2 + 1) \qquad \nabla\ell_n(\eta) = \frac{2(Z_n - \eta)}{(Z_n - \eta)^2 + 1}$$

$$\nabla^2\ell_n(\eta) = \frac{2(Z_n - \eta)^2 - 2}{((Z_n - \eta)^2 + 1)^2} \qquad \nabla^3\ell_n(\eta) = \frac{4((Z_n - \eta)^2 - 3)(\eta - Z_n)}{((\eta - Z_n)^2 + 1)^3},$$

where $Z_n = X_n$. Property (A1) holds by routine interchange of differentiation and integration. Property (A2) holds (for any $\delta > 0$) because $\nabla\ell_n(\eta)$ and $\nabla^2\ell_n(\eta)$ are bounded functions of $\eta$ and $X_n$ jointly. Property (A3) holds (for any neighbourhood of $\eta_0$) because $\nabla^3\ell_n(\eta)$ is a bounded function of $\eta$ and $X_n$ jointly. Property (A4) holds because the pushforward of $\text{Cauchy}(0,1)$ through $\eta(\theta) = \theta^2$ has full support on $\mathbb{R}_+$. In order to verify assumption (A5), suppose there exists a sequence of bounded measurable functions $\phi_r(Z_1, \dots, Z_N) \in [0,1]$ of the data and constants $c, c' > 0$ such that for all $r \to 0$, $r^2 = \omega(\log N/N)$,

$$\mathbb{E}_{\eta_0}\phi_r = O\left(e^{-cNr^2}\right) \quad \text{and} \quad \sup_{\|\eta-\eta_0\|>r}\mathbb{E}_\eta(1 - \phi_r) = O\left(e^{-c'Nr^2}\right).$$

The functions $\phi_r$ are similar to the test functions of Schwartz [55]. Then defining $\mu = \eta\pi$ and $\mu_0 = \eta\pi_0$,

$$\mu(\|\eta - \eta_0\| > r) = \phi_r\mu(\|\eta - \eta_0\| > r) + (1 - \phi_r)\mu(\|\eta - \eta_0\| > r)$$
$$\leq \phi_r + (1 - \phi_r)\mu(\|\eta - \eta_0\| > r)$$
$$= \phi_r + \frac{\int_{\|\eta-\eta_0\|>r}(1 - \phi_r)e^{\ell(\eta)-\ell(\eta_0)}\mu_0}{\int e^{\ell(\eta)-\ell(\eta_0)}\mu_0}.$$

Using the same proof technique as in Theorem 3.3, the denominator satisfies

$$\log\int e^{\ell(\eta)-\ell(\eta_0)}\mu_0(\mathrm{d}\eta) \geq -\frac{d}{2}\log N + O_p(1).$$

By assumption, there exists $c > 0$ such that

$$\mathbb{E}_{\eta_0}[\phi_r] = O\left(e^{-cNr^2}\right) \implies \phi_r = O_p(e^{-cNr^2}),$$

and a $c' > 0$ such that

$$\mathbb{E}_{\eta_0}\left[\int_{\|\eta-\eta_0\|>r}(1 - \phi_r)e^{\ell(\eta)-\ell(\eta_0)}\mu_0\right] = \int_{\|\eta-\eta_0\|>r}\mathbb{E}_\eta(1 - \phi_r)\mu_0$$
$$\leq \sup_{\|\eta-\eta_0\|>r}\mathbb{E}_\eta(1 - \phi_r)$$
$$= O(e^{-cNr^2}) \implies \int_{\|\eta-\eta_0\|>r}(\dots) = O_p\left(e^{-cNr^2}\right).$$

Therefore $\mu(\|\eta - \eta_0\| \geq r) = O_p\left(e^{-cNr^2} + N^{d/2}e^{-c'Nr^2}\right) = O_p\left(e^{(d/2)\log N - c''Nr^2}\right)$; and since $r^2 = \omega(\log N/N)$, $-\log\mu(\|\eta - \eta_0\| \geq r) = \Omega_p(Nr^2)$ as required by (A5). So to complete the proof of (A5) we need to find a suitable $\phi_r$. Fix $\epsilon > 0$, and set

$$\phi_r(Z_1, \dots, Z_N) = \mathbb{1}\left[P_{\eta_0}(|Z - \eta_0| \leq 1) - \frac{1}{N}\sum_{n=1}^N \mathbb{1}[|Z_n - \eta_0| \leq 1] > \epsilon r\right].$$

Under $p_{\eta_0}$, Hoeffding's inequality yields

$$\mathbb{E}_{\eta_0} \phi_r \le e^{-2N\epsilon^2 r^2}.$$

And under $p_\eta$ for $\|\eta - \eta_0\| > r$, for small enough $\epsilon > 0$, $P_{\eta_0}(|Z - \eta_0| \le 1) - P_\eta(|Z - \eta_0| \le 1) \ge 2\epsilon r$. Therefore

$$
\begin{aligned}
\mathbb{E}_\eta[1 - \phi_r(Z_1, \ldots, Z_N)] &= \Pr_\eta\left( P_{\eta_0}(|Z - \eta_0| \le 1) - \frac{1}{N} \sum_{n=1}^N \mathbb{1}[|Z_n - \eta_0| \le 1] \le \epsilon r \right) \\
&\le \Pr_\eta\left( P_\eta(|Z - \eta_0| \le 1) - \frac{1}{N} \sum_{n=1}^N \mathbb{1}[|Z_n - \eta_0| \le 1] \le -\epsilon r \right) \\
&= \Pr_\eta\left( \frac{1}{N} \sum_{n=1}^N \mathbb{1}[|Z_n - \eta_0| \le 1] - P_\eta(|Z - \eta_0| \le 1) \ge \epsilon r \right),
\end{aligned}
$$

at which point we can again apply Hoeffding's inequality, completing the result. $\qquad \square$

**Lemma A.7.** *Fix vectors $u, u_1, \ldots, u_N$ in a separable Hilbert space with inner product denoted $a \cdot b$ and norm denoted $\| \ \|$. Let $v_1, \ldots, v_M$ be drawn from $\{u_1, \ldots, u_N\}$ with probabilities $p_1, \ldots, p_N$ either with or without replacement (if without replacement, the probabilities are renormalized after every draw). Then for all $\epsilon \ge 0$,*

$$
\mathbb{P}\left( \min_{w \ge 0} \left\| \sum_{m=1}^M w_m v_m - u \right\|^2 > \epsilon^{M\left( \frac{q(M,\epsilon)}{2} \right)+1} \|u\|^2 \right) \le e^{-\left( \frac{1 - \log(2)}{2} \right) M},
$$

*where*

$$
q(M, \epsilon) = \mathbb{P}\left( 1 - \max\left\{ 0, \frac{v_M}{\|v_M\|} \cdot \frac{(u - x_{M-1})}{\|u - x_{M-1}\|} \right\}^2 \le \epsilon \right) \qquad x_{M-1} = \operatorname*{arg\,min}_{x \in \mathrm{cone}\{v_1, \ldots, v_{M-1}\}} \|x - u\|^2.
$$

*Proof.* First note that it suffices to analyze the case with replacement, since this case provides an upper bound on the case without replacement. To demonstrate this, we couple two probability spaces—one that draws $v_1, \ldots, v_M$ with replacement, and one without replacement. First, draw an identical vector $v_1$ for both copies. On each subsequent iteration $m > 1$, the "with replacement" copy first draws whether or not it selects a vector that was previously selected by the "without replacement" copy. If it does, it draws that vector independently; if it does not, it selects the same vector as the "without replacement" copy. In any case, at each iteration $m$, the vectors drawn by the "with replacement" copy are always a subset of the vectors drawn by the "without replacement" copy, and hence the minimum over $w \ge 0$ is greater for that copy. It therefore suffices to analyze the case with replacement.

To obtain an upper bound on the probability when sampling with replacement, instead of minimizing over all $w \ge 0$ jointly, suppose we use the following iterative algorithm. Set $x_0 = 0$. At the first iteration, we draw $v_1$ and set the weight $w_1$ by optimizing over $w_1 \ge 0$:

$$
\min_{w_1 > 0} \|w_1 v_1 - u\|^2 = \|u\|^2 \left( 1 - \max\left\{ 0, \frac{v_1 \cdot u}{\|v_1\| \|u\|} \right\}^2 \right).
$$

Set $x_1 = w_1 v_1$, and note that $(u - x_1) \cdot x_1 = 0$. Then at each subsequent iteration $k$, assume the previous iterate is optimized over all nonnegative weights, and hence satisfies $(u - x_{k-1}) \cdot x_{k-1} = 0$. We draw another vector $v_k$, and bound the erorr of the next iterate $x_k$ by optimizing over only the weight $w_k$ for the new vector $v_k$. Then

$$
\begin{aligned}
\|u - x_k\|^2 = \min_{w_1, \ldots, w_k \ge 0} \left\| \sum_{m=1}^k w_m v_m - u \right\|^2 &\le \min_{w_k > 0} \|w_k v_k + x_{k-1} - u\|^2 \\
&= \|u - x_{k-1}\|^2 \left( 1 - \max\left\{ 0, \frac{v_k \cdot (u - x_{k-1})}{\|v_k\| \|u - x_{k-1}\|} \right\}^2 \right).
\end{aligned}
$$

Therefore,

$$
\begin{aligned}
&\mathbb{P}\left( \min_{w \ge 0} \left\| \sum_{m=1}^M w_m v_m - u \right\|^2 \le \epsilon^K \|u\|^2 \right) \\
&\ge \mathbb{P}\left( \text{in at least } K \text{ iterations, } \|x_k - u\|^2 \le \epsilon \|x_{k-1} - u\|^2 \right)
\end{aligned}
$$

$$\geq \mathbb{P}\left( \text{in at least } K \text{ iterations}, 1 - \max\left\{0, \frac{v_k \cdot (u - x_{k-1})}{\|v_k\|\|u - x_{k-1}\|}\right\}^2 \leq \epsilon \right)$$

$$= \sum_{\mathcal{K} \subseteq [M], |\mathcal{K}| \geq K} \mathbb{P}\left( k \in \mathcal{K} \iff 1 - \max\left\{0, \frac{v_k \cdot (u - x_{k-1})}{\|v_k\|\|u - x_{k-1}\|}\right\}^2 \leq \epsilon \right)$$

$$\geq \sum_{\mathcal{K} \subseteq [M], |\mathcal{K}| \geq K} q^k (1 - q)^{M-k}$$

$$= \sum_{k=K}^{M} \binom{M}{k} q^k (1 - q)^{M-k},$$

where

$$q = \mathbb{P}\left( 1 - \max\left\{0, \frac{v_M \cdot (u - x_{M-1})}{\|v_M\|\|u - x_{M-1}\|}\right\}^2 \leq \epsilon \right)$$

$$x_{M-1} = \underset{x \in \text{cone}\{v_1, \ldots, v_{M-1}\}}{\arg\min} \|x - u\|^2$$

So for all $0 \leq K \leq M$,

$$\mathbb{P}\left( \min_{w \geq 0} \left\| \sum_{m=1}^{M} w_m v_m - u \right\|^2 > \epsilon^K \|u\|^2 \right) \leq \text{Binom}(M, K - 1, q).$$

Using the Chernoff bound on the binomial CDF, for all $K - 1 \leq Mq$,

$$\mathbb{P}\left( \min_{w \geq 0} \left\| \sum_{m=1}^{M} w_m v_m - u \right\|^2 > \epsilon^K \|u\|^2 \right) \leq e^{-M\left( \frac{K-1}{M} \log \frac{K-1}{Mq} + (1 - \frac{K-1}{M}) \log \frac{1 - \frac{K-1}{M}}{1-q} \right)}$$

$$= e^{-(K-1) \log \frac{K-1}{Mq} - (M-(K-1)) \log \frac{M-(K-1)}{M(1-q)}}.$$

Substituting $K - 1 = Mq/2$ yields

$$= e^{M((q/2) \log 2 - (1-q/2) \log \frac{1-q/2}{(1-q)})} \leq e^{-\left( \frac{1 - \log(2)}{2} \right) M}.$$

$\square$

*Proof of Corollary 6.1.* Since the potentials are $\beta \text{Cov}_\pi((\ell_n)_{n=1}^N)$ subexponential, Theorem 5.3 guarantees that

$$\forall w \in \mathbb{R}_+^N : 4\beta(w - 1)^T \text{Cov}_\pi((\ell_n)_{n=1}^N)(w - 1) \leq 1, \qquad \overline{\text{KL}}(w) \leq 4\beta(w - 1)^T \text{Cov}_\pi((\ell_n)_{n=1}^N)(w - 1).$$

We apply Lemma A.7 with vectors $\ell_1, \ldots, \ell_N$ (in equivalence classes specified up to a additive constant) and inner product between $\ell_i, \ell_j$ defined by $\text{Cov}_\pi(\ell_i, \ell_j)$. In the notation of Lemma A.7, by assumption, $\|u\|^2 = O_p(N^\alpha)$ and $q(M, \epsilon) = \omega_p(M^{-\rho})$. Substituting $M = (\log N)^{1/(1-\rho)}$, we find that

$$\mathbb{P}\left( 4\beta(w - 1)^T \text{Cov}_\pi((\ell_n)_{n=1}^N)(w - 1) \geq \epsilon^{-\omega_p(\log N) + \alpha \log N} \right) \to 0.$$

Combining this result with the KL bound above yields the final result. $\square$

