# OpenReview forum: "General bounds on the quality of Bayesian coresets"
_NeurIPS.cc/2024/Conference — NeurIPS 2024 poster_

### Official Review · Reviewer_2Qah · 2024-06-30

**Soundness:** 2
**Presentation:** 2
**Contribution:** 2
**Rating:** 4
**Confidence:** 4

**Summary:**

This paper studies the quality of posterior likelihood based on coreset, a subset of the full data. The quality is quantified through KL divergence between the posterior likelihoods based on coreset and full data, so the main theorems demonstrate the upper and lower bounds of KL which are of great concern.

**Strengths:**

The Bayesian coreset approach is interesting and has been studied recently. Upper and lower bounds of the KL divergence are interesting and useful.

**Weaknesses:**

1. Tightness of the upper and lower bounds. I am not sure how different are the bounds from each other. If upper and lower bounds are different from each other, their applications might be limited.

2. There seem to be insufficient experimental results to support the results.

**Questions:**

1. Please describe the gaps between the upper and lower bounds of the KL divergence.

2. The paper seems to be related subsampling in frequent statistics. Is coreset method a Bayesian counterpart?

---

> ### Author Rebuttal · Authors · 2024-08-02
>
> Thanks for your efforts reviewing the manuscript!
>
> > Tightness of the upper and lower bounds. I am not sure how different are the bounds from each other. If upper and lower bounds are different from each other, their applications might be limited.
> > Please describe the gaps between the upper and lower bounds of the KL divergence.
>
> This is a really great question! Reviewer jwKa had the same question, so we duplicate our response here.
>
> The short answer is that the upper and lower bounds aren’t meant to be comparable. They’re tools to show whether an algorithm is working well or poorly. The upper bounds should be used to demonstrate that an algorithm is working well, while the lower bounds should be used to demonstrate that an algorithm is working poorly (much like our usage of our results in the paper in Cor 4.1,4.2,4.3, and 6.1).
>
> For the lower bounds: essentially, you should think of the lower bounds as a “test for poor functioning” of a coreset construction. Roughly, the bounds in Theorem 3.3 and 3.5 say that any reasonable coreset construction algorithm must be good enough to well-approximate the score at the true parameter $g_w/\bar w \approx g/N$. We apply these results in Cor 4.1, 4.2 to show that importance-weighted coresets do not pass the test. If an algorithm passes the test (e.g., $\|g/N - g_w/\bar w\| \to 0$ quickly enough) the lower bounds don’t say much. The really nice thing about the lower bound “test” is that it makes the analysis quite simple: it reduces the problem of understanding minimum KL divergence to just a 2-norm comparison of two vector empirical averages.
>
> For the upper bounds: Theorem 5.3 asserts that as long as you know the coreset construction algorithm is working at least somewhat well (  $(w-1)^TA(w-1) \leq 1$ ), then you can bound the maximum KL divergence. We believe the upper bounds will be relatively tight in this regime, although we have not worked on a proof of this fact. The reason we believe this to be true is that the quadratic expansion of the KL divergence around $w=1$ is roughly $(w-1)^T \mathrm{Cov}(\dots) (w-1)$, which matches Theorem 5.3 with $A = \mathrm{Cov}(\dots)$ and $f(x) = x$, so the gap between the result in Theorem 5.3 and the true KL should be a cubic remainder term that decays quickly.
>
> Note that in our work we have encountered cases where the bounds coincide (ignoring constants), e.g. importance weighted constructions for Gaussian location models, for which both upper and lower bounds yield KL rates of $N/M$; but we believe these cases to be very limited and not of general interest.
>
> We will be sure to include discussion related to this point in the revised manuscript.
>
> > There seem to be insufficient experimental results to support the results.
>
> Please note that this is a theoretical paper; the proofs suffice to support the key contributions of the work. While experimental results are not necessary, we believe that the simulations involving models that violate the conditions of past work are illustrative and add some intuition, support, and clarity to the meaning of the theoretical results.
>
> > The paper seems to be related subsampling in frequent statistics. Is coreset method a Bayesian counterpart?
>
> Yes, it is somewhat related to subsampling methods (e.g., work by HaiYing Wang at UConnecticut and collaborators in recent years). However, building good Bayesian coresets to obtain posterior KL control is a much harder problem in general. In the frequentist work we’ve seen, the goal is to find optimal subsampling weights that minimize some error criterion for point estimates. The lower bounds in our work (Cor 4.1, 4.2) show that *there does not exist any (reasonable) setting* of subsampling weights that yield a good Bayesian coreset; more careful tuning of the weights is required. This is essentially because Bayesian coresets need to approximate the full posterior distribution well, not just a single point estimate.

---

> > ### Comment · Reviewer_2Qah · 2024-08-11
> > **Thanks for response.**
> >
> > KL divergence is a measure to examine the quality of a core set. Small upper bound implies good quality. This is what I agree with the author(s).
> >
> > But I don’t agree with your response that lower bound implies low quality.
> >
> > Lower bound is typically used to justify if the upper bound can be further improved. For instance, if you prove that that
> >
> > 0.1<KL<0.2
> >
> > It is possible that your upper bound 0.2 is not sharp and could be further lowered. Of course it could be also that your lower bound is not sharp. But the lower bound 0.1 doesn’t mean that the core set is bad.

---

> > > ### Author Response · Authors · 2024-08-11
> > > **Response**
> > >
> > > Thanks for your follow up!
> > >
> > > > But I don’t agree with your response that lower bound implies low quality.
> > >
> > > A *large* lower bound does indeed imply low quality. For example, in the case of our Cor 4.1, we prove that the KL must be lower bounded by (ignoring constants) $N/M$ for importance weighting methods; therefore in order to control the KL divergence, importance weighting methods must set $M\propto N$, as otherwise the KL grows without bound as $N\to\infty$. At this point we have shown that if one wants to maintain control of the KL, importance weighted coresets can yield at most a constant reduction in coreset size asymptotically in $N$.
> > >
> > > > Lower bound is typically used to justify if the upper bound can be further improved.
> > >
> > > We are not using lower bounds to prove sharpness of upper bounds in this paper; we are using lower bounds to show that some very popular algorithms perform poorly, and using upper bounds to analyze other algorithms to show that they work well.

---

> > > > ### Comment · Reviewer_EUq5 · 2024-08-12
> > > >
> > > > Thanks. I will keep my (positive) score.

---

> ### Comment · Area_Chair_pnXV · 2024-08-11
> **discussion**
>
> Dear Reviewer 2Qah,
>
> Thank you very much for submitting your review report. The author(s) have posted responses to your review. Could you kindly provide comments on whether your concerns have been adequately addressed?
>
> Best regards, AC

---

### Official Review · Reviewer_EUq5 · 2024-07-04

**Soundness:** 4
**Presentation:** 4
**Contribution:** 3
**Rating:** 7
**Confidence:** 3

**Summary:**

This paper provides new lower and upper asymptotic bounds on the KL divergence between the true posterior and the posterior obtained from common classes of coreset construction algorithms, under milder assumptions than previously used.

My own research is in the area of Bayesian statistics but overall less theoretical. Yet, I could follow along thanks to the authors clear structure and writing. I did not have time to check all the proofs in detail but the authors gave me no reason to disbelieve their results.

**Strengths:**

The paper provides important new insights on the asymptotic performance we can can expect from Bayesian coreset approaches. It is very well written and provides good intuition about their theorems, thus facilitating understanding of the results even for readers you do not have the time or background to go through all the technicalities of the theorems. The two empirical examples are simple but provide good intuition through relating the empirical results with the theory.

**Weaknesses:**

I don’t see any major weaknesses. Of course, it would have been desirable to see the results illustrated on more complicated empirical examples, but I understand why the authors used the examples they did.

**Questions:**

Can the authors provide more intuition on why (close to Eq 4) “The lower threshold ensures that the variance of the importance-weighted log-likelihood is not too large, while the upper threshold ensures sufficient diversity in the draws from subsampling.”?

**Limitations:**

The provided limitations appear sensible to me.

---

> ### Author Rebuttal · Authors · 2024-08-02
>
> Thanks very much for your efforts reviewing the manuscript!
>
> > I don’t see any major weaknesses. Of course, it would have been desirable to see the results illustrated on more complicated empirical examples, but I understand why the authors used the examples they did.
>
> Much appreciated!
>
> > Can the authors provide more intuition on why (close to Eq 4) “The lower threshold ensures that the variance of the importance-weighted log-likelihood is not too large, while the upper threshold ensures sufficient diversity in the draws from subsampling.”?
>
> Great question. At the core, it’s because Cor 4.1 and 4.2 are based on a conditional central limit theorem in Lemma A.3. If the $p_n$ are too small, the variance of the importance-weighted sum (which depends on $1/p_n$) will grow too quickly to satisfy the CLT. If the $p_n$ are too large, the number of unique points drawn will grow too slowly for the empirical average to satisfy the CLT (consider the extreme case where there is one datapoint with $p_n = 1$; the empirical average will just be of a single data point, which of course cannot satisfy a CLT). We will clarify the writing here in the revision.
>
> As a side comment, (A6) probably isn't strictly necessary to achieve the results in Cor 4.1 and 4.2; we could perhaps use another technique not based on the CLT for cases where $p_n$ is asymptotically very small or very large. But we found (A6) to capture all cases of practical interest and so were satisfied with it.

---

### Official Review · Reviewer_vJwU · 2024-07-11

**Soundness:** 4
**Presentation:** 3
**Contribution:** 4
**Rating:** 8
**Confidence:** 3

**Summary:**

The paper derives bounds on the quality of coresets, as measured by forward and reverse KL divergence. The lower bound is used to study importance weighted coresets, leading to the conclusion that importance weighting leads to a large (forward or reverse) KL divergence between the approximate posterior and the posterior unless $\Omega(N)$ points are sampled into the coreset. They also use these results to show that under additional conditions, any coreset must be of size at least $d$, where $d$ is the dimensionality of the parametric class to avoid a large KL divergence. The upper bound is used to show that coresets  of size $O(\log N)$ are sufficient to maintain bounded KL divergences for the subsample-optimize approach.

**Strengths:**

- The corollaries of the main results have consequences for the choice of method for coreset construction that seem insightful.
- Conditions are carefully stated throughout the main text and examples are given to show that the conditions assumed in results do not exclude all interesting cases.

**Weaknesses:**

- Several statements are made in text that seem overly strong. For example, the authors claim that “handling large-scale data in Bayesian inference…”requires exploiting redundancy in the data”. This seems overly strong (without proof). Further, I think there are large-scale data problems in Bayesian inference that can be solved via exploring structure that cannot be solved via exploiting data redundancy. For example, the likelihood in Gaussian process inference with a regularly spaced one-dimensional data can be computed quickly by exploiting structure in the prior/posterior, but I don’t think would be well-addressed by coreset methods (since there is very little data redundancy). Of course, this isn’t really meant as a criticism of coresets, or to suggest this is the type of problem the authors try to solve. I simply think strong claims should be made specific and supported by results.
- (Minor) I find the notation used around measures to be somewhat confusing and I think unorthodox. For example, my understanding is that $\pi_0$ is a measure, but in assumption 3.2, it is also used for a density. Similarly, in assumption 3.2, using $\pi_0(d \theta)$ for the measure itself (without an integral) seems unusual. And the integration notation without a differential element seems unorthodox (I would expect either $d\pi$ or maybe $\pi(d\theta)$). I have also not seen the push forward measure written without either a $\\#$ or $\*$, or more directly as a composition. Perhaps the authors can point me to a standard reference using these notations.
- (Minor) The definition of f in Lemma 3.1 seems unnecessarily complicated. In particular, if $x \leq 1$, f is just 0. Defining f in cases would likely make it more interpretable if space allows.

**Questions:**

- Could the authors provide proof that $L$ smoothness from below implies Lipschitz smoothness? At a minimum it seems differentiability is required in $L$-smoothness. But also it looks to me more like a Lipschitz condition on the derivate of f instead of f itself.
- The authors claim that (2) and (3) satisfy the earlier assumptions, but no reference is provided. I didn’t see a checking of assumptions in the appendix. Could the authors either include this, or point me to the correct section if it is already included?
- Line 191: “Nonuniform probabilities require at least $O(N)$ time and memory…”, should this be $\Omega(N)$? I’m not convinces at least $O(N)$ makes sense as a notion, since $O(N)$ includes, for example, constant time algorithms.
- In section 6, the authors make several references to “exact coresets” without previously introducing the concept. What is meant by this? Does this mean a coreset that results in exactly recovering the posterior, or something else?
- This is certainly beyond the scope of the current work, but I am curious. Are the approximation results presented strong enough to establish contraction for the approximate posterior produced by Bayesian coreset methods? (perhaps following a similar argument to Ray and Szabo, 2020 Theorem 5).

### Reference
Ray and Szabo. Variational Bayes for high-dimensional linear regression with sparse priors. 2020.

**Limitations:**

Limitations seem well-adressed.

### Other Comments:
Consider defining the various big-O etc notation used in a footnote or appendix. These are all reasonably standard, but some come up less than others and I think it would be useful for many readers. Relatedly, $\omega$ and $w$ are quite hard to tell apart. I don’t know if there is a fix for this, but perhaps it is worth considering.

---

> ### Author Rebuttal · Authors · 2024-08-02
>
> First, thank you for your very thorough review!
>
> > Several statements are made in text that seem overly strong. [...]
>
> This is a fair comment. We were thinking mostly of parametric models with conditionally independent data, where competing methods rely on asymptotic normality (either explicitly or implicitly) to achieve scalability. We are happy to adjust the writing to remove unjustified claims.
>
> > (Minor) I find the notation used around measures to be somewhat confusing [...]
>
> Another good point — we were sloppy in a few places. We generally prefer the bare symbol, e.g. $\pi_0$, to denote measures, with an argument $\pi_0(\theta)$ to denote density, and $\int (\dots) \pi_0(\mathrm{d}\theta)$ for integration. We will go through the paper and make sure that the notation is consistent.
>
> For pushforwards, we find the # notation unnecessary; we will stick with just prepending the map. But we will make notation explicitly clear in a new notation section (also explaining asymptotic convergence notation as mentioned elsewhere).
>
> > (Minor) The definition of f in Lemma 3.1 seems unnecessarily complicated. [...]
>
> We agree and will edit this in the revision. As a very minor clarification, note that f is 0 when x >= 1, not <= 1.
>
> > Could the authors provide proof that 𝐿 smoothness from below implies Lipschitz smoothness? [...]
>
> We believe there might be a terminology confusion here. Just in case: Lipschitz smoothness refers to functions $f$ for which $\|\nabla f(x) - \nabla f(y)\| \leq L \|x-y\|$ (for twice differentiable functions, equivalent to $\| \nabla^2 f \|_2 \leq L$). This is different from Lipschitz continuity.
>
> Further, note that the statement in the paper was not that $L$-smoothness below implies Lipschitz smoothness; it was the converse. At Line 145 the paper states that $L$-smoothness below is weaker than Lipschitz smoothness. Specifically, $L$-Lipschitz smoothness implies $L$-smoothness below trivially, because $L$-Lipschitz smoothness implies that the growth of $f$ is no faster than $\frac{L}{2}\|\theta-\theta_0\|^2$ in *either direction for all $\theta_0$*, while $L$-smoothness below only implies a bound on lower growth for a single $\theta_0$.
>
> While responding to this comment we noticed that the statement re: strong concavity is wrong. $L$-smoothness below isn’t weaker than strong concavity, it just doesn’t imply it (or even concavity). The corrected statement should read “$L$-smoothness below is weaker than Lipschitz smoothness, and does not imply concavity”.
>
> > The authors claim that (2) and (3) satisfy the earlier assumptions, but no reference is provided. [...]
>
> You are correct – it was not in the submission. When preparing the manuscript we did verify these properties, but felt the verification was straightforward (these assumptions are routine in the Bayesian asymptotics literature). But you’re right that it should be included, and it will be in the revision. We include a sketch of the verification here:
>
> (A1) There are no issues with interchanging differentiation and integration in either model, and in these cases it is a standard result from statistics that the expected score is 0 and the expected negative Hessian is the expected score outer product.
>
> (A2) We can bound terms both by looking at the expected Frobenius norm with a $(1+\delta)$ power. The Hessian in both models is bounded globally, and hence the Frobenius expectation is finite for any $\delta > 0$.
>
> (A3) The log-likelihood is 3 times differentiable in both models, so the Hessian is locally Lipschitz. Since the Hessians are bounded globally in both models, $R$ can be chosen to be bounded and hence has a finite expectation.
>
> (A4) The priors in both models are twice differentiable and strictly positive everywhere.
>
> (A5) Both models are parametric, identifiable under the $\eta(\theta)$ pushforward, and the likelihoods are continuous in total variation as a function of $\eta$. The condition follows from results from Bayesian asymptotics (e.g. Lemma 10.3 of Asymptotic Statistics by van der Vaart, with conditions of Theorem 10.1 verified using Lemma 10.6).
>
> We will add a full proof of each statement in the appendix, and add commentary in the text for readers about how to prove specifically (A5), which is the only condition that requires more than just routine derivation.
>
> Upon reviewing this we also noticed a typo in (A2) – $\theta_0$ should be $\eta_0$. We will fix this in the revision.
>
> > Line 191: “Nonuniform probabilities require at least 𝑂(𝑁) time and memory…”, should this be Ω(𝑁)? I’m not convinces at least 𝑂(𝑁) makes sense as a notion, since 𝑂(𝑁)includes, for example, constant time algorithms.
>
> Correct! Good catch, thank you; this will be fixed in the revision.
>
> > In section 6, the authors make several references to “exact coresets” without previously introducing the concept. What is meant by this? Does this mean a coreset that results in exactly recovering the posterior, or something else?
>
> Correct, an “exact coreset” is one that recovers the full data posterior. We will clarify this point in the revision.
>
> > This is certainly beyond the scope of the current work, but I am curious. Are the approximation results presented strong enough to establish contraction for the approximate posterior produced by Bayesian coreset methods? [...]
>
> It’s a very interesting question. The general upper bounds presented in this work in Section 5 should be strong enough to achieve this, but the application in Cor 6.1 just demonstrates $KL = O_p(1)$, which isn’t enough. However, one should be able to modify the proof technique for Cor 6.1 to have slightly more stringent conditions to achieve $KL = o_p(1)$, at which point Theorem 5 of Ray & Szabo should be applicable.
>
> > Consider defining the various big-O etc notation used in a footnote or appendix [...]
>
> Agreed; thank you for the suggestion. We will be sure to include a brief definition of each asymptotic notation used in the paper in the revision.

---

> > ### Comment · Reviewer_vJwU · 2024-08-12
> > **Thanks for the comments and clarifications**
> >
> > Thanks to the authors for the detailed response to my comments, especially the clarification on Lipschitz smoothness versus Lipschitz continuity. I think it could be useful for the authors to add a definition for this in the appendix, although I don't think it is essential since this isn't the assumption they are working with directly. The questions I had were answered in detail and I will maintain my rating; I still feel the paper should be accepted.

---

### Official Review · Reviewer_UnLw · 2024-07-12

**Soundness:** 4
**Presentation:** 4
**Contribution:** 4
**Rating:** 7
**Confidence:** 2

**Summary:**

The authors present general upper and lower bounds on the Kullback-Leibler (KL) divergence of coreset approximations. The lower bounds require only mild model assumptions typical of Bayesian asymptotic analyses, while the upper bounds require the log-likelihood functions to satisfy a generalized subexponentiality criterion that is weaker than conditions used in earlier work. The lower bounds are applied to explain the the poor performance of importance sampling-based construction methods. The upper bounds are used to analyze the performance of recent subsample-optimize methods.

**Strengths:**

1. The paper is well-written, and the main technical results are clearly presented.
2. The paper addresses several gaps in the analysis of Bayesian coreset such as providing a theoretical explanation for the previously-observed poor empirical performance of importance sampling-based construction methods through a lower bound on KL divergence.
3. To the best of my knowledge, the theorems are sound and the derivations are accurate.

**Weaknesses:**

The paper offers a theoretical explanation for the suboptimal performance of importance-weighted coreset construction, and the conclusions of Corollary 4.1 and 4.2 are further validated by simulation results (Figure 2).
However, [32] presents extensive experimental results on real-world datasets, which raises questions about the effectiveness of the lower bounds in scenarios where the model is misspecified or when working with real-world data (where models are often misspecified to some extent).

**Questions:**

1. Could you please provide some discussions about whether the theories can be extended to misspecified regime?
2. Providing formal definitions for some notations such as $\Omega_p$ and $\omega_p$ would be helpful.

**Limitations:**

The paper clearly states its limitation.

---

> ### Author Rebuttal · Authors · 2024-08-02
>
> Thank you for reviewing our manuscript! We appreciate your kind words about its clarity, novelty, and significance.
>
> > The paper offers a theoretical explanation for the suboptimal performance of importance-weighted coreset construction [...] however, [the results in] [32] [...] raises questions about the effectiveness of the lower bounds in scenarios where the model is misspecified [...]
> > Could you please provide some discussions about whether the theories can be extended to misspecified regime?
>
> This is a great point to raise! However, we disagree that [32] conflicts with our results. In fact, the results in [32] motivated our hunt for lower bounds to explain the (surprisingly poor) empirical results in that paper.
>
> The experimental results in [32] do not consider scaling in $N$, as all experiments have a fixed value of $N$. So it is not possible to draw conclusions about the validity of the proposed lower bounds—which are all asymptotic in $N$—based on these results. However, the results in Figure 2 of [32] do demonstrate that the importance-weighted coreset construction performs just as poorly as basic uniform subsampling (in terms of posterior approximation error, Poly MMD^3), perhaps with a small constant improvement. This agrees with our Cor 4.1.
>
> The main theoretical result in [32] is Theorem 3.2, which says that $M$ proportional to $\bar m_N / \epsilon^2$ suffices to produce a uniform log-likelihood approximation with $|L - \tilde L| \leq \epsilon |L|$. But note that $|L|$ is the total log-likelihood function, which scales with the number of data $N$ (it is possible to improve this by centering the log-likelihood function so that $|L|$ scales pointwise like $\sqrt{N}$). So $\epsilon$ must be *at most* $O(1/\sqrt{N})$ to keep the KL divergence controlled as $N\to\infty$, and hence $N \propto M$ (since $\bar m_N \sim 1$ per Lemma 3.1 in [32]). This result in [32] is an upper bound; our lower bound proves that one cannot do better than $N\propto M$ using importance-weighting.
>
> Regarding misspecification specifically: note that the goal of a Bayesian coreset construction is to approximate the total log-likelihood function with a weighted subset sum. While the model itself may or may not be useful when misspecified, Bayesian coreset construction (and the theory in this paper) is agnostic to this. Essentially, as long as the data are truly generated conditionally iid from *some* process, the theoretical results in our work should hold for the chosen log-likelihood function, prior, and (possibly unknown) data generating process. But we leave a careful investigation of misspecification for future work. We are happy to mention this as a limitation of the present paper in the revision.
>
> > Providing formal definitions for some notations such as Ω𝑝 and 𝜔𝑝  would be helpful.
>
> Agreed; thank you for the suggestion! We will be sure to include a brief definition of each asymptotic notation used in the paper in the revision.

---

> > ### Comment · Reviewer_UnLw · 2024-08-10
> >
> > Thanks for the clarifying responses to my questions! I am happy to recommend accept.

---

### Official Review · Reviewer_jwKa · 2024-07-15

**Soundness:** 4
**Presentation:** 3
**Contribution:** 3
**Rating:** 7
**Confidence:** 3

**Summary:**

This work contributes to the approximation theory of Bayesian coresets. It establishes asymptotic lower bounds (in KL divergence) of Bayesian coreset approximation that do not require posterior normality assumptions. It also provides an upper bound of the approximation (in KL divergence) when the potentials satisfy the generalized subexponentiality condition. The theories are applied to some recent coreset construction algorithms in the literature to corroborate their observed empirical performances.

**Strengths:**

-  I must preface this review by saying that this paper is not in my expertise. However, as someone unfamiliar with Bayesian corset, I found it well-written, has a good flow, and I believe it presents very solid theoretical results.
- The theoretical results are novel, presenting the first established lower bound for coreset approximation in the literature.
- The upper bound of coreset approximation in this work relaxes the assumptions found in previous studies. I appreciate the author(s) providing examples of subexponential potentials in Propositions A.1 and A.2.

**Weaknesses:**

- Since this manuscript studies both the lower and upper bounds of coreset approximation, a natural question that arises is how well do the lower and upper bounds match each other? Can the author(s) provide an example to see if there is a gap between these two bounds? That is, if we consider a specific set of subexponential potentials with a specific $f$ and $A$, can we show that its upper bound matches its lower bound when $N$ is sufficiently large? Or, are the lower and upper bounds incomparable? If these bounds are optimal, they can guide researchers in designing corset construction algorithms and greatly benefit the community.
- In the second lower bound in Theorem 3.3, is there a trade-off between $N$ and $\|\frac{g}{N} - \frac{g_w}{\bar{w}}\|^2_2$ as $N \to \infty$ while $\|\frac{g}{N} - \frac{g_w}{\bar{w}}\|^2_2 \to 0$? If $\|\frac{g}{N} - \frac{g_w}{\bar{w}}\|^2_2 \to 0$ faster than $N \to \infty$, do we end up having a lower bound that's close to $0$?
- It seems like the role of the sequence $r$ (i.e., how fast does $r \to 1$) is very significant in establishing the lower bound. Specifically, in Corollary 4.1, 4.2, and 4.3, $r$ is carefully chosen as shown in the proofs. I find it necessary to specify these choices of $r$ in the statements of Corollary 4.1, 4.2, and 4.3.

**Questions:**

See weaknesses

**Limitations:**

Yes, the author(s) have addressed the limitations in the conclusion section.

---

> ### Author Rebuttal · Authors · 2024-08-02
>
> First, thank you for your efforts reviewing the manuscript. We're very glad you found it understandable and interesting despite not being in your area!
>
> > Since this manuscript studies both the lower and upper bounds of coreset approximation, a natural question that arises is how well do the lower and upper bounds match each other [...]
>
> This is a really great question! The short answer is that the upper and lower bounds aren’t meant to be comparable. They’re tools to show whether an algorithm is working well or poorly. The upper bounds should be used to demonstrate that an algorithm is working well, while the lower bounds should be used to demonstrate that an algorithm is working poorly (much like the applications of our results in the paper in Cor 4.1,4.2,4.3, and 6.1).
>
> For the lower bounds: essentially, you should think of the lower bounds as a “test for poor functioning” of a coreset construction. Roughly, the bounds in Theorem 3.3 and 3.5 say that any reasonable coreset construction algorithm must be good enough to well-approximate the score at the true parameter $g_w/\bar w \approx g/N$. We apply these results in Cor 4.1, 4.2 to show that importance-weighted coresets do not pass the test. If an algorithm passes the test (e.g., $\|g/N - g_w/\bar w\| \to 0$ quickly enough) the lower bounds don’t say much. The really nice thing about the lower bound “test” is that it makes the analysis quite simple: it reduces the problem of understanding minimum KL divergence to just a 2-norm comparison of two vector empirical averages $g/N$ and $g_w/\bar w$.
>
> For the upper bounds: Theorem 5.3 asserts that as long as you know the coreset construction algorithm is working at least somewhat well (  $(w-1)^TA(w-1) \leq 1$ ), then you can bound the maximum KL divergence. We believe the upper bounds will be relatively tight in this regime, although we have not worked on a proof of this fact. The reason we believe this to be true is that the quadratic Taylor expansion of the KL divergence around $w=1$ is roughly $(w-1)^T \mathrm{Cov}(\dots) (w-1)$, which matches Theorem 5.3 with $A = \mathrm{Cov}(\dots)$ and $f(x) = x$, so the gap between the result in Theorem 5.3 and the true KL should be a cubic remainder term that decays quickly.
>
> Note that in our work we have encountered cases where the bounds do indeed coincide (ignoring constants), e.g. importance weighted constructions for Gaussian location models, for which both upper and lower bounds yield KL rates of $N/M$; but we believe these cases to be very limited and not of general interest.
>
> We will be sure to include discussion related to this point in the revised manuscript.
>
> > In the second lower bound in Theorem 3.3, is there a trade-off between $N$ and $\|g/N - g_w/\bar w\|$ as $N\to\infty$ while $\|g/N - g_w/\bar w\| \to 0$? If $\|g/N - g_w/\bar w\|\to 0$ faster than $N\to\infty$, do we end up having a lower bound that's close to 0?
>
> Correct! Indeed, the lower bounds in Theorem 3.3 are most useful when $\|g/N - g_w/\bar w\|\to 0$ slower than $1/\sqrt{N}$. In this case, the lower bound increases to infinity, which shows that the coreset construction algorithm is not good enough to be useful in practice. If $\|g/N - g_w/\bar w\| \to 0$ faster than that, the lower bound in Theorem 3.3 does not say anything of interest.
>
> > It seems like the role of the sequence $r$ (i.e., how fast does $r\to 1$) is very significant in establishing the lower bound. Specifically, in Corollary 4.1, 4.2, and 4.3, $r$ is carefully chosen as shown in the proofs. I find it necessary to specify these choices of $r$ in the statements of Corollary 4.1, 4.2, and 4.3.
>
> Indeed you are right – choosing $r$ carefully is critical to the proofs. $r$ must be small enough that the quadratic log-likelihood Taylor expansions we use in the proofs are accurate, but large enough that $\pi$ concentrates on $B$ quickly. But the choice of $r$ should not appear in the statements of Cor 4.1, 4.2, and 4.3, as it is not part of the conclusion of these results. Instead, we would be happy to put it as a remark in the text either before / after the corollary statements.

---

> > ### Comment · Reviewer_jwKa · 2024-08-12
> >
> > I am pleased that the authors have thoroughly addressed my questions and concerns. I would like to strongly suggest accepting this paper.

---

### Author Rebuttal · Authors · 2024-08-02

Thank you to all the reviewers for their efforts reviewing our manuscript. We have responded to each reviewer in the comment section below their review. Please let us know if there are any further questions!

---

### Decision · Program_Chairs · 2024-09-25

**Decision:**

Accept (poster)

**Comment:**

The paper addresses the problem of evaluating the quality of Bayesian coreset approximations by deriving new general lower and upper bounds on the Kullback-Leibler (KL) divergence between coreset and full-data posteriors. These bounds are applied to analyze and draw conclusions about the performance of importance-weighted and subsample-optimize coreset methods, with results supported by simulation experiments. The theoretical contributions are strong, particularly the novel lower bounds that avoid strict assumptions. However, concerns were raised about the gap between upper and lower bounds (Reviewers jwKa and 2Qah) and the need for more empirical validation, especially with real-world data (Reviewers UnLw and 2Qah). There is also a suggestion to refine the presentation of technical notations (Reviewer vJwU) and clarify the implications of the lower bounds (Reviewers jwKa, 2Qah, and EUq5).